# Conditional score-based diffusion models for Bayesian inference in infinite dimensions

**Lorenzo Baldassari**
University of Basel
lorenzo.baldassari@unibas.ch

**Ali Siahkoohi**
Rice University
alisk@rice.edu

**Josselin Garnier**
Ecole Polytechnique, IP Paris
josselin.garnier@polytechnique.edu

**Knut Sølna**
University of California Irvine
ksolna@math.uci.edu

**Maarten V. de Hoop**
Rice University
mvd2@rice.edu

## Abstract

Since their initial introduction, score-based diffusion models (SDMs) have been successfully applied to solve a variety of linear inverse problems in finite-dimensional vector spaces due to their ability to efficiently approximate the posterior distribution. However, using SDMs for inverse problems in infinite-dimensional function spaces has only been addressed recently, primarily through methods that learn the unconditional score. While this approach is advantageous for some inverse problems, it is mostly heuristic and involves numerous computationally costly forward operator evaluations during posterior sampling. To address these limitations, we propose a theoretically grounded method for sampling from the posterior of infinite-dimensional Bayesian linear inverse problems based on amortized conditional SDMs. In particular, we prove that one of the most successful approaches for estimating the conditional score in finite dimensions—the conditional denoising estimator—can also be applied in infinite dimensions. A significant part of our analysis is dedicated to demonstrating that extending infinite-dimensional SDMs to the conditional setting requires careful consideration, as the conditional score typically blows up for small times, contrarily to the unconditional score. We conclude by presenting stylized and large-scale numerical examples that validate our approach, offer additional insights, and demonstrate that our method enables large-scale, discretization-invariant Bayesian inference.

## 1 Introduction

Inverse problems seek to estimate unknown parameters using noisy observations or measurements. One of the main challenges is that they are often ill-posed. A problem is ill-posed if there are no solutions, or there are many (two or more) solutions, or the solution is unstable in relation to small errors in the observations [1]. A common approach to transform the original ill-posed problem into a well-posed one is to formulate it as a least-squares optimization problem that minimizes the difference between observed and predicted data. However, minimization of the data misfit alone negatively impacts the quality of the obtained solution due to the presence of noise in the data and the inherent nullspace of the forward operator [2, 3]. Casting the inverse problem into a Bayesian probabilistic framework allows, instead, for a full characterization of all the possible solutions [4–6]. The Bayesian approach consists of putting a prior probability distribution describing uncertainty in the parameters of interest, and finding the posterior distribution over these parameters [7]. The prior must be chosen appropriately in order to mitigate the ill-posedness of the problem and facilitate computation of the posterior. By adopting the Bayesian formulation, rather than finding one single solution to the inverse

37th Conference on Neural Information Processing Systems (NeurIPS 2023).

problem (e.g., the maximum a posteriori estimator [8]), a distribution of solutions—the posterior—is finally obtained, whose samples are consistent with the observed data. The posterior distribution can then be sampled to extract statistical information that allows for uncertainty quantification [9].

Over the past few years, deep learning-based methods have been successfully applied to analyze linear inverse problems in a Bayesian fashion. In particular, recently introduced score-based diffusion models (SDMs) [10] have become increasingly popular, due to their ability of producing approximating samples from the posterior distribution [11, 12]. An SDM consists of a diffusion process, which gradually perturbs the data distribution toward a tractable distribution according to a prescribed stochastic differential equation (SDE) by progressively injecting Gaussian noise, and a generative model, which entails a denoising process defined by approximating the time-reversal of the diffusion. Crucially, the denoising stage is also a diffusion process [13] whose drift depends on the logarithmic gradients of the noised data densities—the scores—which are estimated by Song et al. [10] using a neural network. Among the advantages of SDMs over other deep generative models is that they produce high-quality samples, matching the performance of generative adversarial networks [14], without suffering from training instabilities and mode-collapse [10, 15]. Additionally, SDMs are not restricted to invertible architectures like normalizing flows [16], which often limits the complexity of the distributions that can be learned. Finally, and most importantly to the scope of this work, SDMs have demonstrated superior performance in a variety of inverse problems, such as image inpainting [10, 17], image colorization [10], compressing sensing, and medical imaging [12, 18].

In the aforementioned cases, SDMs have been applied by assuming that the data distribution of interest is supported on a finite-dimensional vector space. However, in many inverse problems, especially those governed by partial differential equations (PDEs), the unknown parameters to be estimated are functions (e.g., coefficient functions, boundary and initial conditions, or source functions) that exist in a suitable function space, typically an infinite-dimensional Hilbert space. The inverse heat equation or the elliptic inverse source problem presented in [19] are typical examples of ill-posed inverse problems that are naturally formulated in infinite-dimensional Hilbert spaces. In addition to these PDE-based examples, other interesting cases that are not PDE-based include geometric inverse problems (e.g., determining the Riemann metric from geodesic information or the background velocity map from travel time information in geophysics [20]) and inverse problems involving singular integral operators [21]. A potential solution for all of these problems could be to discretize the input and output functions into finite-dimensional vectors and apply SDMs to sample from the posterior. However, theoretical studies of current diffusion models suggest that performance guarantees do not generalize well on increasing dimensions [22–24]. This is precisely why Andrew Stuart's guiding principle to study a Bayesian inverse problem for functions—"avoid discretization until the last possible moment" [5]—is critical to the use of SDMs.

Motivated by Stuart's principle, in this work we *define a conditional score in the infinite-dimensional setting*, a critical step for studying Bayesian inverse problems directly in function spaces through SDMs. In particular, we show that using this newly defined score as a reverse drift of the diffusion process yields a generative stage that samples, under specified conditions, from the correct target conditional distribution. We carry out the analysis by focusing on two cases: the case of a Gaussian prior measure and the case of a general class of priors given as a density with respect to a Gaussian measure. Studying the model for a Gaussian prior measure provides illuminating insight, not only because it yields an analytic formula of the score, but also because it gives a full characterization of SDMs in the infinite-dimensional setting, showing under which conditions we are sampling from the correct target conditional distribution and how fast the reverse SDE converges to it. It also serves as a guide for the analysis in the case of a general class of prior measures. Finally, we conclude this work by presenting, in Section 6, stylized and large-scale numerical examples that demonstrate the applicability of our SDM. Specifically, we show that our SDM model (i) is able to *approximate non-Gaussian multi-modal distributions*, a challenging task that poses difficulties for many generative models [25]; (ii) is *discretization-invariant*, a property that is a consequence of our theoretical and computational framework being built on the infinite-dimensional formulation proposed by Stuart [5]; and (iii) is *applicable to solve large-scale Bayesian inverse problems*, which we demonstrate by applying it to a large-scale problem in geophysics, i.e., the linearized wave-equation-based imaging via the Born approximation that involves estimating a 256×256-dimensional unknown parameter.

**Related works**    Our work is primarily motivated by Andrew Stuart's comprehensive mathematical theory for studying PDE-governed inverse problems in a Bayesian fashion [5]. In particular, we are

interested in the infinite-dimensional analysis [7, 26], which emphasizes the importance of analyzing PDE-governed inverse problems directly in function space before discretization.

Our paper builds upon a rich and ever expanding body of theoretical and applied works dedicated to SDMs. Song et al. [10] defined SDMs integrating both score-based (Hyvärinen [27]; Song and Ermon [17]) and diffusion (Sohl-Dickstein et al. [28]; Ho et al. [29]) models into a single continuous-time framework based on stochastic differential equations. The generative stage in SDMs is based on a result from Anderson [13] proving that the denoising process is also a diffusion process whose drift depends on the scores. This result holds only in *vector spaces*, which explains the difficulties to extend SDMs to more general function spaces. Initially, there have been attempts to project the input functions into a finite-dimensional feature space and then apply SDMs (Dupont et al. [30]; Phillips et al. [31]). However, these approaches *are not discretization-invariant*. It is only very recently that SDMs have been directly studied in function spaces, specifically infinite-dimensional Hilbert spaces. Kerrigan et al. [32] generalized diffusion models to operate directly in function spaces, but they did not consider the time-continuous limit based on SDEs (Song et al. [10]). Dutordoir et al. [33] proposed a denoising diffusion generative model for performing Bayesian inference of functions. Lim et al. [34] generalized score matching for trace-class noise corruptions that live in the Hilbert space of the data. However, as Kerrigan et al. [32] and Dutordoir et al. [33], they did not investigate the connection to the forward and backward SDEs as Song et al. [10] did in finite dimensions. Three recent works, Pidstrigach et al. [24], Franzese et al. [35] and Lim et al. [36], finally established such connection for the *unconditional setting*. In particular, Franzese et al. [35] used results from infinite-dimensional SDEs theory (Föllmer and Wakolbinger [37]; Millet et al. [38]) close to Anderson [13].

Among the mentioned works, Pidstrigach et al. [24] is the closest to ours. We adopt their formalism to establish theoretical guarantees for sampling from the conditional distribution. Another crucial contribution comes from Batzolis et al. [39], as we build upon their proof to show that the score can be estimated by using a denoising score matching objective conditioned on the observed data [17, 40]. A key element in Pidstrigach et al. [24], emphasized also in our analysis, is obtaining an estimate on the expected square norm of the score that needs to be *uniform in time*. We explicitly compute the expected square norm of the conditional score in the case of a Gaussian prior measure, which shows that a uniform in time estimate is *not always possible in the conditional setting*. This is not surprising, given that the singularity in the conditional score as noise vanishes is a well-known phenomenon in finite dimensions and has been investigated in many works, both from a theoretical and a practical standpoint [41, 42]. In our paper, we provide a set of concrete conditions to be satisfied to ensure a uniform estimate in time for a general class of prior measures in infinite dimensions.

Pidstrigach et al. [24] have also proposed a method for performing conditional sampling, building upon the approach introduced by Song et al. [12] in a finite-dimensional setting. Like our approach, their method can be viewed as a contribution to the literature on likelihood-free, simulation-based inference [43, 44]. Specifically, the algorithm proposed by Pidstrigach et al. [24] relies on a projection-type approach that incorporates the observed data into the unconditional sampling process via a proximal optimization step to generate intermediate samples consistent with the measuring acquisition process. This allows Pidstrigach et al. [24] *to avoid defining the conditional score*[1]. While their method has been shown to work well with specific inverse problems, such as medical imaging [12], it is primarily heuristic, and its computational efficiency varies depending on the specific inverse problem at hand. Notably, their algorithm may require numerous computationally costly forward operator evaluations during posterior sampling. Furthermore, their implementation does not fully exploit the discretization-invariance property achieved by studying the problem in infinite dimensions since they employ a UNet to parametrize their score, limiting the evaluation of their score function to the training interval. The novelty of our work is then twofold. First, we provide theoretically grounded guarantees for an approach that is not heuristic and can be implemented such that *it is not constrained to the grid on which we trained our network*. As a result, we show that we effectively take advantage of the discretization-invariance property achieved by adopting the infinite-dimensional

---

[1]We note that, in a newer version of their paper submitted to arXiv on October 3, 2023 (four months after our submission to arXiv and NeurIPS 2023), Pidstrigach et al. [24] abandoned the projection-type approach. Instead, they invoke the conditional score function to perform posterior sampling and solve an inverse problem in a similar fashion to ours (we refer to Section 8.3 of their paper for details). However, they still do not address the well-posedness of the forward-reverse conditional SDE and the singularity of the conditional score, and their implementation is based on UNets and, thus, is not discretization-invariant.

formulation proposed by Stuart [5]. Second, we perform discretization-invariant Bayesian inference by learning an *amortized version of the conditional score*. This is done by making the score function depending on the observations. As a result, provided that we have access to high-quality training data, during sampling we can input any new observation that we wish to condition on directly during simulation of the reverse SDE. In this sense, our method is *data-driven*, as the information about the forward model is implicitly encoded in the data pairs used to learn the conditional score. This addresses a critical gap in the existing literature, as the other approach using infinite-dimensional SDM resorts to projections onto the measurement subspace for sampling from the posterior—a method that not only lacks theoretical interpretation but may also yield unsatisfactory performance due to costly forward operator computations. There are well-documented instances in the literature where amortized methods can be a preferred option in Bayesian inverse problems [45–50], as they reduce inference computational costs by incurring an offline initial training cost for a deep neural network that is capable of approximating the posterior for unseen observed data, provided that one has access to a set of data pairs that adequately represent the underlying joint distribution.

**Main contributions**  The main contribution of this work is the *analysis of conditional SDMs in infinite-dimensional Hilbert spaces*. More specifically,

- We introduce the conditional score in an infinite-dimensional setting (Section 3).
- We provide a comprehensive analysis of the forward-reverse conditional SDE framework in the case of a *Gaussian prior measure*. We explicitly compute the expected square norm of the conditional score, which shows that a uniform in time estimate *is not always possible for the conditional score*. We prove that as long as we start from the invariant distribution of the diffusion process, the reverse SDE converges to the target distribution exponentially fast (Section 4).
- We provide a set of conditions to be satisfied to ensure a uniform in time estimate for a general class of prior measures that *are given as a density with respect to a Gaussian measure*. Under these conditions, the conditional score—used as a reverse drift of the diffusion process in SDMs—yields a generative stage that samples from the target conditional distribution (Section 5).
- We prove that the conditional score can be estimated via a conditional denoising score matching objective in infinite dimensions (Section 5).
- We present examples that validate our approach, offer additional insights, and demonstrate that our method enables *large-scale*, *discretization-invariant* Bayesian inference (Section 6).

## 2  Background

Here, we review the definition of unconditional score-based diffusion models (SDMs) in infinite-dimensional Hilbert spaces proposed by Pidstrigach et al. [24], as we will adopt the same formalism to define SDMs for conditional settings. We refer to Appendix A for a brief introduction to key tools of probability theory in function spaces.

Let $\mu_{\text{data}}$ be the target measure, supported on a separable Hilbert space $(H, \langle \cdot, \cdot \rangle)$. Consider a forward infinite-dimensional diffusion process $(X_t)_{t \in [0,T]}$ for continuous time variable $t \in [0, T]$, where $X_0$ is the starting variable and $X_t$ its perturbation at time $t$. The diffusion process is defined by the following SDE:

$$dX_t = -\frac{1}{2} X_t dt + \sqrt{C} dW_t, \tag{1}$$

where $C : H \to H$ is a fixed trace class, positive-definite, symmetric covariance operator and $W_t$ is a Wiener process on $H$. Here and throughout the paper, the initial conditions and the driving Wiener processes in (1) are assumed independent.

The forward SDE evolves $X_0 \sim \mu_0$ towards the Gaussian measure $\mathcal{N}(0, C)$ as $t \to \infty$. The goal of score-based diffusion models is to convert the SDE in (1) to a generative model by first sampling $X_T \sim \mathcal{N}(0, C)$, and then running the correspondent reverse-time SDE. In the finite-dimensional case, Song et al. [10] show that the reverse-time SDE requires the knowledge of the score function $\nabla \log p_t(X_t)$, where $p_t(X_t)$ is the density of the marginal distribution of $X_t$ (from now on denoted $\mathbb{P}_t$) with respect to the Lebesgue measure. In infinite-dimensional Hilbert spaces, there is no natural analogue of the Lebesgue measure (for additional details, see [51]) and the density is thus no longer well defined. However, Pidstrigach et al. [24, Lemma 1] notice that, in the finite-dimensional setting

where $H = \mathbb{R}^D$, the score can be expressed as follows:

$$C\nabla_x \log p_t(x) = -(1 - e^{-t})^{-1}\left(x - e^{-t/2}\mathbb{E}[X_0|X_t = x]\right),\tag{2}$$

for $t > 0$. Since the right-hand side of the expression above is also well-defined in infinite dimensions, Pidstrigach et al. [24] formally define the score as follows:

**Definition 1.** *In the infinite-dimensional setting, the score or reverse drift is defined by*

$$S(t,x) := -(1 - e^{-t})^{-1}\left(x - e^{-t/2}\mathbb{E}[X_0|X_t = x]\right).\tag{3}$$

Assuming that the expected square norm of the score is uniformly bounded in time, Pidstrigach et al. [24, Theorem 1] shows that the following SDE

$$dZ_t = \frac{1}{2}Z_t dt + S(T - t, Z_t)\mathrm{d}t + \sqrt{C}dW_t, \qquad Z_0 \sim \mathbb{P}_T,\tag{4}$$

is the time-reversal of (1) and the distribution of $Z_T$ is thus equal to $\mu_0$, proving that the forward-reverse SDE framework of Song et al. [10] generalizes to the infinite-dimensional setting. The reverse SDE requires the knowledge of this newly defined score, and one approach for estimating it is, similarly to [10], by using the denoising score matching loss [40]

$$\mathbb{E}\left[\left\|\widetilde{S}(t, X_t) + (1 - e^{-t})^{-1}(X_t - e^{-t/2}X_0)\right\|^2\right],\tag{5}$$

where $\widetilde{S}(t, X_t)$ is typically approximated by training a neural network.

## 3 The conditional score in infinite dimensions

Analogous to the score function relative to the unconditional SDM in infinite dimensions, we now define the score corresponding to the reverse drift of an SDE when conditioned on observations. We consider a setting where $X_0$ is an $H$-valued random variable and $H$ is an infinite-dimensional Hilbert space. Denote by

$$Y = AX_0 + B,\tag{6}$$

a noisy observation given by $n$ linear measurements, where the measurement acquisition process is represented by a linear operator $A : H \to \mathbb{R}^n$, and $B \sim \mathcal{N}(0, C_B)$ represents the noise, with $C_B$ a $n \times n$ nonnegative matrix. Within a Bayesian probabilistic framework, solving (6) amounts to putting an appropriately chosen prior probability distribution $\mu_0$ on $X_0$, and sampling from the conditional distribution of $X_0$ given $Y = y$.

To the best of our knowledge, the only existing algorithm which performs conditional sampling using infinite-dimensional diffusion models on Hilbert spaces is based on the work of Song et al. [12]. The idea, adapted to infinite dimensions by Pidstrigach et al. [24], is to incorporate the observations into the unconditional sampling process of the SDM via a proximal optimization step to generate intermediate samples that are consistent with the measuring acquisition process. Our method relies instead on *utilizing the score of infinite-dimensional SDMs conditioned on observed data*, which we introduce in this work. We begin by defining the conditional score, by first noticing that, in finite dimensions, we have the following lemma:

**Lemma 1.** *In the finite-dimensional setting where $H = \mathbb{R}^D$, we can express the conditional score function for $t > 0$ as*

$$C\nabla_x \log p_t(x|y) = -(1 - e^{-t})^{-1}\left(x - e^{-t/2}\mathbb{E}\left[X_0|Y = y, X_t = x\right]\right).\tag{7}$$

Since the right-hand side of (7) is well-defined in infinite dimensions, by following the same line of thought of Pidstrigach et al. [24] we formally define the score as follows:

**Definition 2.** *In the infinite-dimensional setting, the conditional score is defined by*

$$S(t,x,y) := -(1 - e^{-t})^{-1}\left(x - e^{-t/2}\mathbb{E}\left[X_0|Y = y, X_t = x\right]\right).\tag{8}$$

**Remark 1.** *It is possible to define the conditional score in infinite-dimensional Hilbert spaces by resorting to the results of [37, 38], see Appendix C.1.*

For Definition 2 to make sense, we need to show that if we use (8) as the drift of the time-reversal of the SDE in (1) conditioned on $y$, then it will sample the correct conditional distribution of $X_0$ given $Y = y$ in infinite dimensions. In the next sections, we will carry out the analysis by focusing on two cases: the case of a Gaussian prior measure $\mathcal{N}(0, C_\mu)$, and the case where the prior of $X_0$ is given as a density with respect to a Gaussian measure, i.e.,

$$X_0 \sim \mu_0, \qquad \frac{d\mu_0}{d\mu}(x_0) = \frac{e^{-\Phi(x_0)}}{\mathbb{E}_\mu[e^{-\Phi(X_0)}]}, \qquad \mu = \mathcal{N}(0, C_\mu), \tag{9}$$

where $C_\mu$ is positive and trace class and $\Phi$ is bounded with $\mathbb{E}_\mu[\|C_\mu \nabla_H \Phi(X_0)\|^2] < +\infty$.

## 4  Forward-reverse conditional SDE framework for a Gaussian prior measure

We begin our analysis of the forward-reverse conditional SDE framework by examining the case where the prior of $X_0$ is a Gaussian measure. This case provides illuminating insight, not only because it is possible to get an analytic formula of the score, but also because it offers a full characterization of SDMs in the infinite-dimensional setting, showing under which conditions we are sampling from the correct target conditional distribution and how fast the reverse SDE converges to it. We also show that the conditional score can have a singular behavior at small times when the observations are noiseless, in contrast with the unconditional score under similar hypotheses.

We assume that $\Phi = 0$ in (9). All distributions in play are Gaussian:

$$X_0 \sim \mathcal{N}(0, C_\mu), \tag{10}$$

$$X_t | X_0 \sim \mathcal{N}(e^{-t/2} X_0, (1 - e^{-t}) C), \tag{11}$$

$$X_0 | Y \sim \mathcal{N}(M_o Y, C_o), \tag{12}$$

$$X_t | Y \sim \mathcal{N}(e^{-t/2} M_o Y, e^{-t} C_o + (1 - e^{-t}) C), \tag{13}$$

where $M_o = C_\mu A^* (A C_\mu A^* + C_B)^{-1}$ and $C_o = C_\mu - C_\mu A^* (A C_\mu A^* + C_B)^{-1} A C_\mu$. By Mercer theorem [52], there exist $(\mu_j)$ in $[0, +\infty)$ and an orthonormal basis $(v_j)$ in $H$ such that $C_\mu v_j = \mu_j v_j$ $\forall j$. We consider the infinite-dimensional case with $\mu_j > 0$ $\forall j$. We assume that $C_\mu$ is trace class so that $\sum_j \mu_j < +\infty$. We assume that the functions $v_j$ are eigenfunctions of $C$ and we denote by $\lambda_j$ the corresponding eigenvalues.

We assume an observational model corresponding to observing a finite-dimensional subspace of $H$ spanned by $v_{\eta(1)}, \ldots, v_{\eta(n)}$ corresponding to $g_k = v_{\eta(k)}$, $k = 1, \ldots, n$, where $g_j \in H$ is such that $(Af)_j = \langle g_j, f \rangle$. We denote $\mathcal{I}^{(n)} = \{\eta(1), \ldots, \eta(n)\}$. We assume moreover $C_B = \sigma_B^2 I_n$. Let $Z_t$ be the solution of reverse-time SDE:

$$dZ_t = \frac{1}{2} Z_t dt + S(T - t, Z_t, y) dt + \sqrt{C} dW_t, \quad Z_0 \sim X_T | Y = y. \tag{14}$$

We want to show that the reverse SDE we have just formulated in (14) indeed constitutes a reversal of the stochastic dynamics from the forward SDE in (1) conditioned on $y$. To this aim, we will need the following lemma:

**Lemma 2.** *We define $Z^{(j)} = \langle v_j, Z \rangle$, $p^{(j)} = \lambda_j / \mu_j$ for all $j$. We also define $y^{(j)} = y_{\eta(j)}$ for $j \in \mathcal{I}^{(n)}$ and $y^{(j)} = 0$ otherwise, and $q^{(j)} = \mu_j / \sigma_B^2$ for $j \in \mathcal{I}^{(n)}$ and $q^{(j)} = 0$ otherwise. Then we can write for all $j$*

$$dZ_t^{(j)} = \mu^{(x,j)}(T - t) Z_t^{(j)} dt + \mu^{(y,j)}(T - t) y^{(j)} dt + \sqrt{\lambda_j} dW^{(j)}, \tag{15}$$

*with $W^{(j)}$ independent and identically distributed standard Brownian motions,*

$$\mu^{(x,j)}(t) = \frac{1}{2} - \frac{e^t p^{(j)} (1 + q^{(j)})}{1 + (e^t - 1) p^{(j)} (1 + q^{(j)})}, \qquad \mu^{(y,j)}(t) = \frac{e^{t/2} p^{(j)} q^{(j)}}{1 + (e^t - 1) p^{(j)} (1 + q^{(j)})}. \tag{16}$$

*Proof.* The proof is a Gaussian calculation. It relies on computing $\langle S, v_j \rangle$, which yields an analytic formula. See Appendix B. □

Lemma 2 enables us to discuss when we are sampling from the correct target conditional distribution $X_0 | Y \sim \mathcal{N}(M_o Y, C_o)$. We can make a few remarks:

- In the limit $T \to \infty$, we get $\mu^{(x,j)}(T-t) \to -1/2$ and $\mu^{(y,j)}(T-t) \to 0$.

- If $j \notin \mathcal{I}^{(n)}$ then we have the same mode dynamics as in the unconditional case. Thus we sample from the correct target distribution if $T$ is large or if we start from $Z_0^{(j)} \sim \mathcal{N}(0, \Sigma_0^{(j)})$ for $\Sigma_0^{(j)} = \mu_j e^{-T} + \lambda_j(1 - e^{-T})$, which is the distribution of $X_T^{(j)} = \langle X_T, v_j \rangle$ given $Y = y$.

- If $j \in \mathcal{I}^{(n)}$ and we start from $Z_0^{(j)} \sim \mathcal{N}(\bar{z}_0^{(j)}, \Sigma_0^{(j)})$, then we find $Z_T^{(j)} \sim \mathcal{N}(\bar{z}_T^{(j)}, \Sigma_T^{(j)})$ with

$$\Sigma_T^{(j)} = \Sigma_0^{(j)} \left( \frac{e^T}{(1 + (e^T - 1)p^{(j)}(1 + q^{(j)}))^2} \right) + \frac{\mu_j}{1 + q^{(j)}} \left( 1 - \frac{1}{1 + (e^T - 1)p^{(j)}(1 + q^{(j)})} \right),$$
(17)

$$\bar{z}_T^{(j)} = \frac{\bar{z}_0^{(j)} e^{T/2}}{1 + (e^T - 1)p^{(j)}(1 + q^{(j)})} + \frac{y^{(j)} q^{(j)}}{1 + q^{(j)}} \left( 1 - \frac{1}{1 + (e^T - 1)p^{(j)}(1 + q^{(j)})} \right).$$
(18)

  The distribution of $X_0^{(j)} = \langle X_0, v_j \rangle$ given $Y = y$ is $\mathcal{N}(y^{(j)} q^{(j)}/(1 + q^{(j)}), \mu_j/(1 + q^{(j)}))$. As $\bar{z}_T^{(j)} \to y^{(j)} q^{(j)}/(1 + q^{(j)})$ and $\Sigma_T^{(j)} \to \mu_j/(1 + q^{(j)})$ as $T \to +\infty$, this shows that we sample from the exact target distribution (the one of $X_0$ given $Y = y$) for $T$ large.

- If we start the reverse-time SDE from the correct model

$$\bar{z}_0^{(j)} = \frac{e^{-T/2} y^{(j)} q^{(j)}}{1 + q^{(j)}}, \qquad \Sigma_0^{(j)} = \frac{\mu_j e^{-T}}{1 + q^{(j)}} + \lambda_j(1 - e^{-T}),$$
(19)

  then indeed $Z_T^{(j)} \sim \mathcal{N}(y^{(j)} q^{(j)}/(1 + q^{(j)}), \mu_j/(1 + q^{(j)}))$. This shows that, for any $T$, $Z_T$ has the same distribution as $X_0$ given $Y = y$, which is the exact target distribution. We can show similarly that $Z_{T-t}$ has the same distribution as $X_t$ given $Y = y$ for any $t \in [0, T]$.

- In the case that $\sigma_B = 0$ so that we observe the mode values perfectly for $j \in \mathcal{I}^{(n)}$, then

$$\mu^{(x,j)}(t) = \frac{1}{2} - \frac{e^t}{e^t - 1}, \qquad \mu^{(y,j)}(t) = \frac{e^{t/2}}{e^t - 1},$$
(20)

  and indeed $\lim_{t \uparrow T} Z_t^{(j)} = y^{(j)}$ a.s. Indeed the $t^{-1}$ singularity at the origin drives the process to the origin like in the Brownian bridge.

Our analysis shows that, as long as we start from the invariant distribution of the diffusion process, we are able to sample from the correct target conditional distribution and that happens exponentially fast. This proves that the score of Definition 2 is the reverse drift of the SDE in (14). Additionally, the analysis shows that the score is uniformly bounded, except when there is no noise in the observations, blowing up near $t = 0$.

**Remark 2.** *Note that, for $q^{(j)} = 0$, we obtain the unconditional model:*

$$dZ_t^{(j)} = \mu^{(j)}(T-t) Z_t^{(j)} dt + \sqrt{\lambda_j} dW^{(j)}, \text{ with } \mu^{(j)}(t) = \frac{1}{2} - \frac{e^t p^{(j)}}{1 + (e^t - 1)p^{(j)}}.$$
(21)

*If $C = C_\mu$, the square expectation of the norm and the Lipschitz constant of the score are uniformly bounded in time:* $\sup_{j, t \in [0,T]} |\mu^{(j)}(t)| = 1/2$.

**Proposition 1.** *The score is $S(t, x, y) = \sum_j S_G^{(j)}(t, \langle x, v_j \rangle, y^{(j)}) v_j$, $S_G^{(j)}(t, x^{(j)}, y^{(j)}) = \left( \mu^{(x,j)}(T-t) - 1/2 \right) x^{(j)} + \mu^{(y,j)}(T-t) y^{(j)}$ and it satisfies*

$$\mathbb{E}[\|S(t, X_t, y)\|_H^2 | Y = y] = \sum_j \frac{e^t(1 + q^{(j)})}{1 + (e^t - 1)p^{(j)}(1 + q^{(j)})} \frac{\lambda_j^2}{\mu_j}.$$
(22)

*Proof.* The proof is a Gaussian calculation given in Appendix B. $\qquad \square$

In the unconditional setting, we have $\mathbb{E}[\|S(t, X_t)\|_H^2] = \sum_j \frac{e^t}{1 + (e^t - 1)p^{(j)}} \frac{\lambda_j^2}{\mu_j}$ which is equal to $\sum_j \lambda_j$ when $C = C_\mu$. It is indeed uniformly bounded in time.

In the conditional and noiseless setting ($\sigma_B = 0$), we have $\mathbb{E}[\|S(t, X_t, y)\|_H^2 | Y = y] = \sum_{j \notin \mathcal{I}^{(n)}} \frac{e^t}{1 + (e^t - 1)p^{(j)}} \frac{\lambda_j^2}{\mu_j} + \sum_{j \in \mathcal{I}^{(n)}} \frac{\lambda_j}{1 - e^{-t}}$, which blows up as $1/t$ as $t \to 0$. This result shows that the extension of the score-based diffusion models to the conditional setting is not trivial.

# 5 Well-posedness for the reverse SDE for a general class of prior measures

We are now ready to consider the case of a general class of prior measures given as a density with respect to a Gaussian measure. The analysis of this case resembles the one of Pidstrigach et al. [24] for the unconditional setting. The main challenge is the singularity of the score for small times, an event that in the Gaussian case was observed in the noiseless setting. In this section we will provide a set of conditions to be satisfied by $\Phi$ in (9), so that the conditional score is bounded uniformly in time. The existence of this bound is needed to make sense of the forward-reverse conditional SDE, and to prove the accuracy and stability of the conditional sampling.

We start the analysis by recalling that, in the infinite-dimensional case, the conditional score is (8). It is easy to get a first estimate:

$$\mathbb{E}[\|S(t, X_t, y)\|_H^2 | Y = y] \leq (1 - e^{-t})^{-1} \mathrm{Tr}(C). \tag{23}$$

The proof follows from Jensen inequality and the law of total expectation, see Appendix C. Note that (23) is indeed an upper bound of (22) since $\mathrm{Tr}(C) = \sum_j \lambda_j$.

Note that the bound (23) is also valid for the unconditional score $S(t, x) = -(1 - e^{-t})^{-1}\big(x - e^{-t/2}\mathbb{E}[X_0 | X_t = x]\big)$. We can observe that the upper bound (23) blows up in the limit of small times. We can make a few comments:

- The bound (23) is convenient for positive times, but the use of Jensen's inequality results in a very crude bound for small times. As shown in the previous section, we know that there exists a bound (21) for the unconditional score in the Gaussian case that is uniform in time.
- The singular behavior as $1/t$ at small time $t$ is, however, not artificial. Such a behavior is needed in order to drive the state to the deterministic initial condition when there are exact observations. This behavior has been exhibited by (20) and (22) in the Gaussian case when $\sigma_B = 0$. This indicates that the following assumption (24) is not trivial in the conditional setting.

For Definition 2 to make sense in the more general case where the prior of $X_0$ is given as a density with respect to a Gaussian measure, we will need to make the following assumption.

**Assumption 1.** *For any $y \in \mathbb{R}^n$, we have*

$$\sup_{t \in [0,T]} \mathbb{E}\big[\|S(t, X_t, y)\|_H^2 | Y = y\big] < \infty. \tag{24}$$

We are now ready to state the analogous result to Pidstrigach et al. [24, Theorem 1].

**Proposition 2.** *Under Assumption 1, the solution of the reverse-time SDE*

$$dZ_t = \frac{1}{2} Z_t dt + S(T - t, Z_t, y) dt + \sqrt{C} dW_t, \qquad Z_0 \sim X_T | Y = y, \tag{25}$$

*satisfies $Z_T \sim X_0 | Y = y$.*

*Proof.* Given Assumption 1, the proof follows the same steps as the one given in [24] for the unconditional score. See Appendix C for the full proof. $\square$

Assumption 1 is satisfied under some appropriate conditions. In the following proposition, we provide a set of conditions that ensure the satisfaction of this assumption. It shows that it is possible to get an upper bound in (23) that is uniform in time provided some additional conditions are fulfilled.

**Proposition 3.** *We assume that $C_\mu$ in (9) and $C$ in (1) have the same basis of eigenfunctions $(v_j)$ and we define $X_t^{(j)} = \langle X_t, v_j \rangle$ and $S^{(j)}(t, x, y) = \langle S(t, x, y), v_j \rangle$ so that in (1) $S(t, x, y) = \sum_j S^{(j)}(t, x, y) v_j$. We assume an observational model as described in Section 4 and that the $p^{(j)}(1 + q^{(j)})$ are uniformly bounded with respect to $j$ and that $C$ is of trace class. We make a modified version of assumption in (9) as follows. We assume that 1) the conditional distribution of $X_0$ given $Y = y$ is absolutely continuous with respect to the Gaussian measure $\mu$ with a Radon-Nikodym derivative proportional to $\exp(-\Psi(x_0, y))$; 2) we have $\Psi(x_0, y) = \sum_j \Psi^{(j)}(x_0^{(j)}, y)$, $x_0^{(j)} = \langle x_0, v_j \rangle$; 3) for $\psi^{(j)}(x^{(j)}, y) = \exp(-\Psi^{(j)}(x^{(j)}, y))$ we have*

$$\frac{1}{K} \leq |\psi^{(j)}(x^{(j)}, y)| \leq K, \qquad |\psi^{(j)}(x^{(j)}, y) - \psi^{(j)}(x^{(j)'}, y)| \leq L|x^{(j)'} - x^{(j)}|, \tag{26}$$

*where $K$ and $L$ do not depend on $j$. Then Assumption 1 holds true.*

*Proof.* The proof is given in Appendix C. □

To use the new score function of Definition 2 for sampling from the posterior, we need to define a way to estimate it. In other words, we need to define a loss function over which the difference between the true score and a neural network $s_\theta(t, x_t, y)$ is minimized in $\theta$. A natural choice for the loss function is

$$\mathbb{E}_{t \sim U(0,T), x_t, y \sim \mathcal{L}(X_t, Y)}\left[\|S(t, x_t, y) - s_\theta(t, x_t, y)\|_H^2\right], \tag{27}$$

however it cannot be minimized directly since we do not have access to the ground truth conditional score $S(t, x_t, y)$. Therefore, in practice, a different objective has to be used. Batzolis et al. [39] proved that, in finite dimensions, a denoising score matching loss can be used:

$$\mathbb{E}_{t \sim U(0,T), x_0, y \sim \mathcal{L}(X_0, Y), x_t \sim \mathcal{L}(X_t | X_0 = x_0)}\left[\|C\nabla_{x_t} \ln p(x_t | x_0) - s_\theta(t, x_t, y)\|^2\right]. \tag{28}$$

This expression involves only $\nabla_{x_t} \log p(x_t | x_0)$ which can be computed analytically from the transition kernel of the forward diffusion process, also in infinite dimensions. In the following proposition, we build on the arguments of Batzolis et al. [39] and provide a proof that the conditional denoising estimator is a consistent estimator of the conditional score in infinite dimensions.

**Proposition 4.** *Under Assumption 1, the minimizer in $\theta$ of*

$$\mathbb{E}_{x_0, y \sim \mathcal{L}(X_0, Y), x_t \sim \mathcal{L}(X_t | X_0 = x_0)}\left[\| -(1 - e^{-t})^{-1}(x_t - e^{-t/2}x_0) - s_\theta(t, x_t, y)\|_H^2\right] \tag{29}$$

*is the same as the minimizer of*

$$\mathbb{E}_{x_t, y \sim \mathcal{L}(X_t, Y)}\left[\|S(t, x_t, y) - s_\theta(t, x_t, y)\|_H^2\right]. \tag{30}$$

*The same result holds if we add $t \sim \mathcal{U}(0, T)$ in the expectations.*

*Proof.* The proof combines some of the arguments of Batzolis et al. [39] and steps of the proof of Lemma 2 in [24], see Appendix C. □

**Remark 3.** *A statement of robustness can be written as in [24, Theorem 2].*

## 6   Numerical experiments

To put the presented theoretical results into practice, we provide two examples. The first stylized example aims at showcasing (i) the ability of our method in capturing nontrivial conditional distributions; and (ii) the discretization-invariance property of the learned conditional SDM. In the second example, we sample from the posterior distribution of a linearized seismic imaging problem in order to demonstrate the applicability of our method to large-scale problems. In both examples, in order to enable learning in function spaces, we parameterize the conditional score using Fourier neural operators [53]. Details regarding our experiment and implementation[2] are presented at Appendix D.

**Stylized example**   Inspired by Phillips et al. [31], we define the target density via the relation $x_0 = ay^2 + \varepsilon$ with $\varepsilon \sim \Gamma(1, 2)$, $a \sim \mathcal{U}\{-1, 1\}$, and $y \in [-3, 3]$. Figure 1a illustrates the samples $x_0$ evaluated on a fine $y$ grid. After training (details in Appendix D), we sample the conditional distribution on uniformly sampled grids between $[-3, 3]$, each having 20 to 40 grid points. Figures 1b and 1c show the predicted samples for grid sizes of 25 and 35, respectively. The marginal conditionals associated with $y = -1.0, 0.0, 0.5$ are shown in Figures 1d–1f, respectively. The gray shaded density in the bottom row of Figure 1 indicates the ground truth density, and colored estimated densities correspond to different discretizations of the horizontal axis. The visual inspection of samples and estimated densities indicates that our approach is indeed discretization-invariant.

**Linearized seismic imaging example**   In this experiment, we address the problem of estimating the short-wavelength component of the Earth's subsurface squared-slowness model (i.e., seismic image; cf. Figure 2a) given surface measurements and a long-wavelength, smooth squared-slowness model (cf. Figure 2b). Following Orozco et al. [54], in order to reduce the high dimensionality of surface measurements, we apply the adjoint of the forward operator, the Born scattering operator, to the measurements and use the outcome (cf. Figure 2c) instead of measured data to condition the SDM. After training, given previously unseen observed data, we use the SDM to sample $10^3$ posterior

---

[2]Code to reproduce results can be found at https://github.com/alisiahkoohi/csgm.

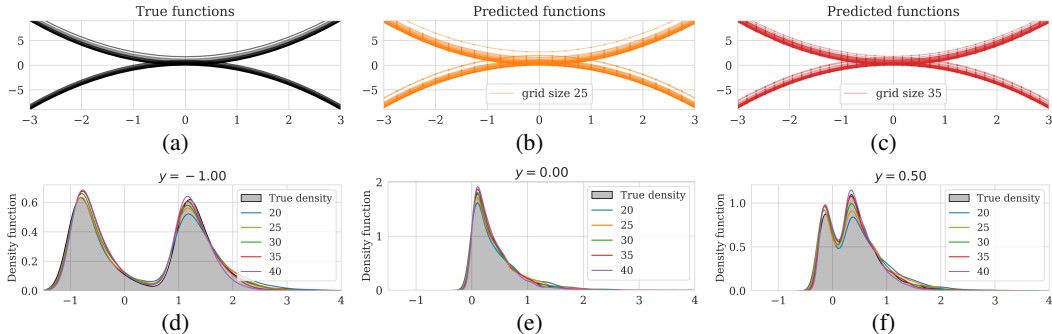

Figure 1: A depiction of the method's discretization invariance. Top row displays ground truth (a) and predicted samples (functions) on a uniformly sampled grid with (b) 25 and (c) 35 grid points. Bottom row shows conditional distribution marginals for (d) $y = -1.0$, (e) $y = 0.0$, and (f) $y = 0.5$.

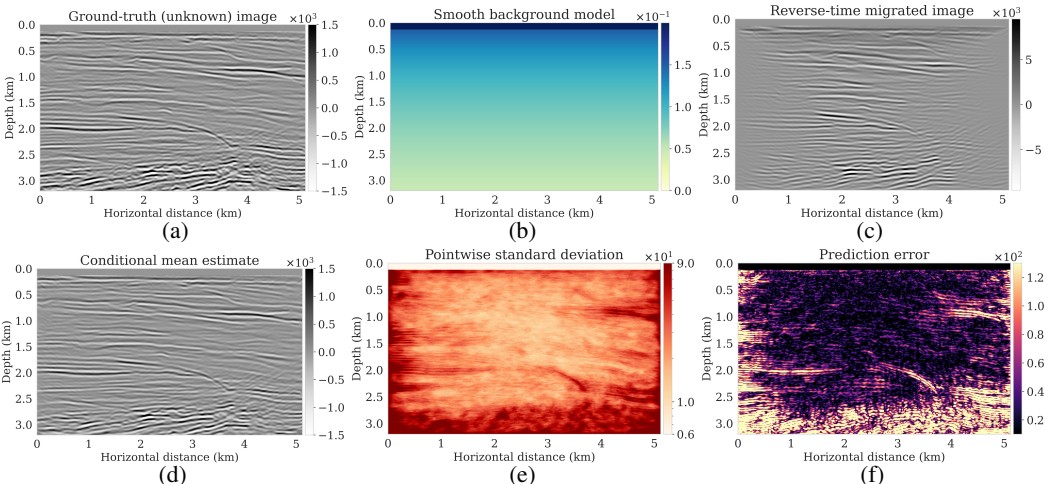

Figure 2: Seismic imaging and uncertainty quantification. (a) Ground-truth seismic image. (b) Background squared-slowness. (c) Data after applying the adjoint Born operator. (d) Conditional (posterior) mean. (e) Pointwise standard deviation. (f) Absolute error between Figures 2a and 2d.

samples to estimate the conditional mean (cf. Figure 2d), which corresponds to the minimum-variance estimate [55], and the pointwise standard deviation (cf. Figure 2e), which we use to quantify the uncertainty. As expected, the pointwise standard deviation highlights areas of high uncertainty, particularly in regions with complex geological structures—such as near intricate reflectors and areas with limited illumination (deep and close to boundaries). We also observe a strong correlation between the pointwise standard deviation and the error in the conditional mean estimate (Figure 2f), confirming the accuracy of our Bayesian inference method.

## 7  Conclusions

We introduced a theoretically-grounded method that *is able to perform conditional sampling in infinite-dimensional Hilbert (function) spaces using score-based diffusion models*. This is a foundational step in using diffusion models to perform Bayesian inference. To achieve this, we learned the infinite-dimensional score function, as defined by Pidstrigach et al. [24], conditioned on the observed data. Under mild assumptions on the prior, this newly defined score—used as the reverse drift of the diffusion process—yields a generative model that samples from the posterior of a linear inverse problem. In particular, the well-known singularity in the conditional score for small times can be avoided. Building on these results, we presented stylized and large-scale examples that showcase the validity of our method and its discretization-invariance, a property that is a consequence of our theoretical and computational framework being built on infinite-dimensional spaces.

## Acknowledgments

JG was supported by Agence de l'Innovation de Défense – AID - via Centre Interdisciplinaire d'Etudes pour la Défense et la Sécurité – CIEDS - (project 2021 - PRODIPO). LB, AS, and MVdH acknowledge support from the Simons Foundation under the MATH + X program, the Department of Energy under grant DE-SC0020345, and the corporate members of the Geo-Mathematical Imaging Group at Rice University. KS was supported by Air Force Office of Scientific Research under grant FA9550-22-1-0176 and the National Science Foundation under grant DMS-2308389.

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

# A  Probability measures on infinite-dimensional Hilbert spaces

In this section, we briefly present some fundamental notions related to probability measures on infinite-dimensional spaces, specifically separable Hilbert spaces $(H, \langle \cdot, \cdot \rangle)$. There is abundant literature on the subject. For more details we refer to Stuart [5], Pidstrigach et al. [24], Kerrigan et al. [32], Prato [51] and references therein.

## A.1  Gaussian measures on Hilbert spaces

**Definition 3.** *Let $(\Omega, \mathcal{F}, \mathbb{P})$ be a probability space. A measurable function $X : \Omega \to H$ is called a Gaussian random element (GRE) if for any $h \in H$, the random variable $\langle h, X \rangle$ has a scalar Gaussian distribution.*

Every GRE $X$ has a mean element $m \in H$ defined by

$$m = \int_\Omega X(\omega) d\mathbb{P}(\omega),$$

and a linear covariance operator $C : H \to H$ defined by

$$Ch = \int_\Omega \langle h, X(\omega) \rangle X(\omega) d\mathbb{P}(\omega) - \langle m, h \rangle m, \quad \forall h \in H.$$

We denote $X \sim \mathcal{N}(m, C)$ for a GRE in $H$ with mean element $m$ and covariance operator $C$. It can be shown that the covariance operator of a GRE is trace class, positive-definite and symmetric. Conversely, for any trace class, positive-definite and symmetric linear operator $C : H \to H$ and every $m \in H$, there exists a GRE with $X \sim \mathcal{N}(m, C)$. This leads us to the following definition:

**Definition 4.** *If $X$ is a GRE, the pushforward of $\mathbb{P}$ through $X$, denoted by $\mathbb{P}_X$, is called a Gaussian probability measure on $H$. We will write $\mathbb{P}_X = \mathcal{N}(m, C)$.*

Let $X \sim \mathcal{N}(m, C)$. We can make a few remarks:
1) For any $h \in H$, we have $\langle h, X \rangle \sim \mathcal{N}(\langle h, m \rangle, \langle Ch, h \rangle)$.
2) $C$ is compact. By Mercer theorem [52] there exists $(\lambda_j)$ and an orthonormal basis of eigenfunctions $(v_j)$ such that $\lambda_j \geq 0$ and $Cv_j = \lambda_j v_j \forall j$. We consider the infinite-dimensional case in which $\lambda_j > 0$ $\forall j$.
3) Suppose $m = 0$ (we call the Gaussian measure of $X$ centered). The expected square norm of $X$ is given by

$$\mathbb{E}[\|X\|_H^2] = \mathbb{E}\left[ \sum_{j=1}^\infty \langle v_j, X \rangle^2 \right] = \sum_{j=1}^\infty \langle Cv_j, v_j \rangle = \sum_{j=1}^\infty \lambda_j = \text{Tr}(C),$$

which is finite since $C$ is trace class.

## A.2  Absolutely continuous measures and the Feldman-Hajek theorem

Here we introduce the notion of absolute continuity for measures.

**Definition 5.** *Let $\mu$ and $\nu$ be two probability measures on $H$ equipped with its Borel $\sigma$-algebra $\mathcal{B}(H)$. Measure $\mu$ is absolutely continuous with respect to $\nu$ (we write $\mu \ll \nu$) if $\mu(\Sigma) = 0$ for all $\Sigma \in \mathcal{B}(H)$ such that $\nu(\Sigma) = 0$.*

**Definition 6.** *If $\mu \ll \nu$ and $\nu \ll \mu$ then $\mu$ and $\nu$ are said to be equivalent and we write $\mu \sim \nu$. If $\mu$ and $\nu$ are concentrated on disjoint sets then they are called singular; in this case we write $\mu \perp \nu$.*

Another notion that will be used throughout the paper is the Radon-Nikodym derivative.

**Theorem 1.** *Let $\mu$ and $\nu$ be two measures on $(H, \mathcal{B}(H))$ and $\nu$ be $\sigma$-finite. If $\mu \ll \nu$, then there exists a $\nu$-measurable function $f$ on $H$ such that*

$$\mu(A') = \int_{A'} f d\nu, \quad \forall A' \in \mathcal{B}(H).$$

*Furthermore, $f$ is unique $\nu$-a.e. and is called the Radon-Nikodym derivative of $\mu$ with respect to $\nu$. It is denoted by $d\mu/d\nu$.*

**Remark 4.** *In the paper, we will sometimes refer to $f$ as the density of $\mu$ with respect to $\nu$.*

We are finally able to state the Feldman-Hajek theorem in its general form.

**Theorem 2.** *The following statements hold.*

1. *Gaussian measures $\mu = \mathcal{N}(m_1, C_1)$, $\nu = \mathcal{N}(m_2, C_2)$ are either singular or equivalent.*

2. *They are equivalent if and only if the following conditions hold:*

   (i) *$\nu$ and $\mu$ have the same Cameron-Martin space $H_0 = C_1^{1/2}(H) = C_2^{1/2}(H)$.*
   (ii) *$m_1 - m_2 \in H_0$.*
   (iii) *The operator $(C_1^{-1/2} C_2^{1/2})(C_1^{-1/2} C_2^{1/2})^* - I$ is a Hilbert-Schmidt operator on the closure $\overline{H_0}$.*

3. *If $\mu$ and $\nu$ are equivalent and $C_1 = C_2 = C$, then $\nu$-a.s. the Radon-Nikodym derivative $d\mu/d\nu$ is given by*
$$\frac{d\mu}{d\nu}(h) = e^{\Psi(h)},$$
   *where $\Psi(h) = \langle C^{-1/2}(m_1 - m_2), C^{-1/2}(h - m_2)\rangle - \frac{1}{2}\|C^{-1/2}(m_1 - m_2)\|_H^2 \, \forall h \in H$.*

### A.3 Bayes' theorem for inverse problems

Let $H$ and $K$ be separable Hilbert spaces, equipped with the Borel $\sigma$-algebra, and $A : H \to K$ a measurable mapping. We want to solve the inverse problem of finding $X$ from $Y$, where
$$Y = A(X) + B$$
and $B \in K$ denotes the noise. We adopt a Bayesian approach to this problem. We let $(X, Y) \in H \times K$ be a random variable and compute $X|Y$. We first specify $(X, Y)$ as follows:
1) Prior: $X \sim \mu_0$ measure on $H$.
2) Noise: $B \sim \eta_0$ measure on $K$, with $B$ independent from $X$.

The random variable $Y|X$ is then distributed according to the measure $\eta_x$, the translate of $\eta_0$ by $A(X)$. We assume that $\eta_x \ll \eta_0$. Thus for some potential $\Psi : H \times K \to \mathbb{R}$,
$$\frac{d\eta_x}{d\eta_0}(y) = e^{-\Psi(x,y)}.$$

The potential $\Psi(\cdot, y)$ satisfying the above formula is often termed the negative log likelihood of the problem. Now define $\nu_0$ to be the product measure $\nu_0 = \mu_0 \times \eta_0$. We can finally state the following infinite-dimensional analogue of the Bayes' theorem.

**Theorem 3.** *Assume that $\Psi : H \times K \to \mathbb{R}$ is $\nu_0$-measurable and define*
$$Z(y) = \int e^{-\Psi(x,y)} d\mu_0.$$

*Then*
$$\frac{d\mu_0(\cdot|Y = y)}{d\mu_0}(x) = \frac{1}{Z(y)} e^{-\Psi(x,y)},$$
*where $\mu_0(\cdot|Y = y)$ is the conditional distribution of $X$ given $Y = y$.*

## B  Proofs of Section 4

### B.1  Proofs of Lemma 2 and Proposition 1

We assume that $C_\mu$ in (9) and $C$ in (1) have the same basis of eigenfunctions $(v_j)$ and that $C_\mu v_j = \mu_j v_j$, $C v_j = \lambda_j v_j \, \forall j$. We define $X_t^{(j)} = \langle X_t, v_j\rangle$, $y^{(j)} = \langle y, v_j\rangle$ and $S^{(j)}(t, x, y) = \langle S(t, x, y), v_j\rangle$ so that in (1) $S(t, x, y) = \sum_j S^{(j)}(t, x, y)v_j$. We assume $j \in \mathcal{I}^{(n)}$ so that we consider a mode corresponding to an observation. We then have
$$dX_t^{(j)} = -\frac{1}{2}X_t^{(j)}dt + \sqrt{\lambda^{(j)}}dW_t^{(j)},$$

with $W^{(j)}$ standard Brownian motions which are independent for the different modes $j$. Note also that with $C_\mu$ and $C$ having the same basis of eigenfunctions the system of modes is diagonalized so that the $X_t^{(j)}$ processes are independent with respect to mode $j$, both for the observed and un-observed modes. Thus we have

$$X_0^{(j)} = \sqrt{\mu_j}\eta_0^{(j)}, \quad Y^{(j)} = X_0^{(j)} + \sigma_B\eta_1^{(j)}, \quad X_t^{(j)} = X_0^{(j)}e^{-t/2} + \sqrt{\lambda_j(1 - e^{-t})}\eta_2^{(j)},$$

for $\eta_i^{(j)}$ independent standard Gaussian random variables. We then seek

$$x_0^{(j,y)} = \mathbb{E}[X_0^{(j)} \mid X_t = x, Y = y] = \mathbb{E}[X_0^{(j)} \mid X_t^{(j)} = x^{(j)}, Y^{(j)} = y^{(j)}],$$

which in this Gaussian setting is the $L^2$ projection of $X_0^{(j)}$ onto $X_t^{(j)}$ and $Y^{(j)}$. Thus we can write $x_0^{(j,y)} = ax^{(j)} + by^{(j)}$ with $(a, b)$ solving

$$\mathbb{E}[(aX_t^{(j)} + bY^{(j)} - X_0^{(j)})Y^{(j)}] = 0, \quad \mathbb{E}[(aX_t^{(j)} + bY^{(j)} - X_0^{(j)})X_t^{(j)}] = 0,$$

which gives

$$a = \frac{e^{t/2}}{1 + (e^t - 1)p^{(j)}(1 + q^{(j)})}, \qquad b = \frac{p^{(j)}q^{(j)}(e^t - 1)}{1 + (e^t - 1)p^{(j)}(1 + q^{(j)})},$$

for $p^{(j)} = \lambda_j/\mu_j$, $q^{(j)} = \mu_j/\sigma_B^2$.

We then get in view of (8)

$$S^{(j)}(t, x, y) = -\left(\frac{e^t p^{(j)}(1 + q^{(j)})}{1 + (e^t - 1)p^{(j)}(1 + q^{(j)})}\right)x^{(j)} + \left(\frac{e^{t/2}p^{(j)}q^{(j)}}{1 + (e^t - 1)p^{(j)}(1 + q^{(j)})}\right)y^{(j)}. \quad (31)$$

Note that with some abuse of notation we then have

$$S^{(j)}(t, x, y) = S^{(j)}(t, x^{(j)}, y^{(j)}),$$

which is important since then also the time reversed system diagonalizes. We remark that for an unobserved mode we get by a similar, but easier, calculation

$$S^{(j)}(t, x, y) = -\left(\frac{e^t p^{(j)}}{1 + (e^t - 1)p^{(j)}}\right)x^{(j)},$$

which simply corresponds to setting $\sigma_B = \infty$ in (31).

Consider next $\mathbb{E}[S^{(j)}(t, x, y)^2]$. Note first that

$$\begin{aligned}
\mathbb{E}[X_t^{(j)} \mid Y = y] &= \left(\frac{q^{(j)}e^{-t/2}}{1 + q^{(j)}}\right)y^{(j)}, \\
\mathrm{Var}[X_t^{(j)} \mid Y = y] &= e^{-t}\mathrm{Var}[X_t^{(j)} \mid Y = y] + \lambda_j(1 - e^{-t}) \\
&= (1 + (e^t - 1)p^{(j)}(1 + q^{(j)}))\left(\frac{\mu_j e^{-t}}{1 + q^{(j)}}\right).
\end{aligned}$$

We can then easily check that the score is conditionally centered $\mathbb{E}[S^{(j)}(t, X_t, Y) \mid Y = y] = 0$ and we then get

$$\begin{aligned}
\mathbb{E}[(S^{(j)}(t, X_t, Y))^2 \mid Y = y] &= \left(\frac{e^t p^{(j)}(1 + q^{(j)})}{1 + (e^t - 1)p^{(j)}(1 + q^{(j)})}\right)^2 \mathrm{Var}[X_t^{(j)} \mid Y = y] \\
&= \frac{e^t \mu_j(p^{(j)})^2(1 + q^{(j)})}{1 + (e^t - 1)p^{(j)}(1 + q^{(j)})},
\end{aligned}$$

which gives Proposition 1 upon summing over the mode index $j$, where we define $q^{(j)} = 0$ for the unobserved modes.

# C Proofs of Section 5

## C.1 Discussion about an alternative approach

The following lemma is a complementary result related to Remark 1. It shows that we can actually derive the expression of the score from the results contained in Millet et al. [38]. The result is powerful, but requires the verification of technical conditions.

**Lemma 3.** *Under the conditions stated in Proposition 3 the score is defined by*

$$S(x,y,t) = -(1-e^{-t})^{-1}\left(x - e^{-t/2}\mathbb{E}\left[X_0|X_t = x, Y = y\right]\right) \tag{32}$$

*and the time reversed diffusion takes the form in (14).*

*Proof.* Define

$$X^{(j)} = \langle v_j, X\rangle \text{ for } Cv_j = \lambda_j v_j.$$

Then

$$dX^{(j)} = -\frac{1}{2}X^{(j)}dt + \sqrt{\lambda_j}dW^{(j)} \text{ for } W^{(j)} = \langle v_j, W\rangle, \tag{33}$$

and where we assume that $C$ is of trace class. This is then an infinite dimensional system of the type considered in Millet et al. [38]. We proceed to verify some conditions stated in Millet et al. [38]: (i) the coefficients of the system (33) satisfy standard growth and Lipschitz continuity conditions (assumption $(H1, H4)$ satisfied); (ii) the coefficients depend on finitely many coordinates (assumption $(H2)$ satisfied); the system is time independent and diagonal (assumption $(H5)$ satisfied). Moreover define $\check{x}^{(j)} = (x_1, \ldots, x_{j-1}, x_{j+1}, \ldots)$, then the law of $X_t^{(j)}$ given $\check{X}_t^{(j)}$ has for $t > 0$ density $p_t(x^{(j)}|\check{X}_t^{(j)} = \check{x}^{(j)}, Y = y)$ with respect to Lebesgue measure and so that for $t_0 > 0$ and each $j$: $\int_{t_0}^{T}\mathbb{E}[|\partial_{x^{(j)}}\log(p_t(x^{(j)})|Y = y)dt < \infty$. Then it follows from Theorems 3.1 and 4.3 in Millet et al. [38] that the time reversed problem is associated with the well-posed martingale problem defined by the coefficients in (14) for the score being:

$$\langle v_j, S(x,y,t)\rangle = \frac{\lambda_j\frac{\partial}{\partial x^j}\left(p_t(x^{(j)}|\check{X}_t^{(j)} = \check{x}^{(j)}, Y = y)\right)}{p_t(x^{(j)}|\check{X}_t^{(j)} = \check{x}^{(j)}, Y = y)},$$

with the convention that the right hand side is null on the set $\{p_t(x^{(j)}|\check{X}_t^{(j)} = \check{x}^{(j)}, Y = y) = 0\}$.

It then follows for $t > 0$

$$\langle v_j, S(x,y,t)\rangle$$

$$= \int_{\mathbb{R}}d\mu_0(x_0^{(j)}|\check{X}_t^{(j)} = \check{x}^{(j)}, Y = y)\frac{\lambda_j\frac{\partial}{\partial x^j}\left(p_t(x^{(j)}|\check{X}_t^{(j)} = \check{x}^{(j)}, X_0^{(j)} = x_0^{(j)}, Y = y)\right)}{p_t(x^{(j)}|\check{X}_t^{(j)} = \check{x}^{(j)}, Y = y)}.$$

We then get

$$\langle v_j, S(x,y,t)\rangle = -\int_{\mathbb{R}}d\mu_0(x_0^{(j)}|\check{X}_t^{(j)} = \check{x}^{(j)}, Y = y)$$

$$\times \left(\frac{x^j - e^{-t/2}x_0^{(j)}}{1 - e^{-t}}\right)\frac{\left(p_t(x^{(j)}|\check{X}_t^{(j)} = \check{x}^{(j)}, X_0^{(j)} = x_0^{(j)}, Y = y)\right)}{p_t(x^{(j)}|\check{X}_t^{(j)} = \check{x}^{(j)}, Y = y)}$$

$$= -\int_{\mathbb{R}}d\mu_0(x_0^{(j)}|X_t = x, Y = y)\left(\frac{x^j - e^{-t/2}x_0^{(j)}}{1 - e^{-t}}\right).$$

$\square$

## C.2 A preliminary lemma

The following lemma is the equivalent of [24, Lemma 3]. It is used in the forthcoming proof of Proposition 2.

**Lemma 4.** *In the finite-dimensional setting $x \in \mathbb{R}^D$, we have for any $0 \le s \le t \le T$:*

$$\nabla \log p_{t,y}(x_t) = e^{(t-s)/2} \mathbb{E}_y \big[ \nabla \log p_{s,y}(X_s) | X_t = x_t \big],$$

*where $\mathbb{E}_y$ is the expectation with respect to the distribution of $X_0$ and $W$ given $Y = y$ and $p_{t,y}$ is the pdf of $X_t$ under this distribution.*

*Proof.* We can write

$$p_{t,y}(x_t) = \int_{\mathbb{R}^D} p_{s,y}(x_s) p_{t|s,y}(x_t|x_s) dx_s,$$

where $p_{t|s,y}(\cdot|x_s)$ is the pdf of $X_t$ given $Y = y$ and $X_s = x_s$. It is, in fact, equal to the pdf of $X_t$ given $X_s = x_s$, which is the pdf of the multivariate Gaussian distribution with mean $\exp(-(t-s)/2)x_s$ and covariance $(1 - \exp(-(t-s)))C$. Therefore

$$p_{t,y}(x_t) = \int_{\mathbb{R}^D} p_{s,y}(x_s) p_{t|s}(x_t|x_s) dx_s.$$

We can then deduce that

$$\nabla p_{t,y}(x_t) = \frac{1}{(2\pi)^{D/2}(1 - e^{-(t-s)})^{1/2} \det(C)^{1/2}} \int_{\mathbb{R}^D} dx_s p_{s,y}(x_s)$$
$$\times \nabla_{x_t} \exp\Big( -\frac{1}{2(1 - e^{-(t-s)})}(x_t - \exp(-(t-s)/2)x_s)^T C^{-1}(x_t - \exp(-(t-s)/2)x_s) \Big)$$

$$= -\frac{e^{(t-s)/2}}{(2\pi)^{D/2}(1 - e^{-(t-s)})^{1/2} \det(C)^{1/2}} \int_{\mathbb{R}^D} dx_s p_{s,y}(x_s)$$
$$\times \nabla_{x_s} \exp\Big( -\frac{1}{2(1 - e^{-(t-s)})}(x_t - \exp(-(t-s)/2)x_s)^T C^{-1}(x_t - \exp(-(t-s)/2)x_s) \Big)$$

$$= \frac{e^{(t-s)/2}}{(2\pi)^{D/2}(1 - e^{-(t-s)})^{1/2} \det(C)^{1/2}} \int_{\mathbb{R}^D} dx_s \nabla_{x_s} \big( p_{s,y}(x_s) \big)$$
$$\times \exp\Big( -\frac{1}{2(1 - e^{-(t-s)})}(x_t - \exp(-(t-s)/2)x_s)^T C^{-1}(x_t - \exp(-(t-s)/2)x_s) \Big)$$

$$= e^{(t-s)/2} \int_{\mathbb{R}^D} dx_s \nabla_{x_s} \big( p_{s,y}(x_s) \big) p_{t|s}(x_t|x_s),$$

which gives

$$\nabla p_{t,y}(x_t) = e^{(t-s)/2} \int_{\mathbb{R}^D} p_{t|s}(x_t|x_s) p_{s,y}(x_s) \nabla \log p_{s,y}(x_s) dx_s.$$

Using again that $p_{t|s,y}(\cdot|x_s) = p_{t|s}(\cdot|x_s)$ and $p_{t|s,y}(x_t|x_s) = \frac{p_{(s,t),y}(x_s,x_t)}{p_{s,y}(x_s)}$, we get

$$\nabla p_{t,y}(x_t) = e^{(t-s)/2} \int_{\mathbb{R}^D} p_{t|s,y}(x_t|x_s) p_{s,y}(x_s) \nabla \log p_{s,y}(x_s) dx_s$$
$$= e^{(t-s)/2} \int_{\mathbb{R}^D} p_{(s,t),y}(x_s, x_t) \nabla \log p_{s,y}(x_s) dx_s.$$

Since $\nabla \log p_{t,y}(x_t) = \frac{\nabla p_{t,y}(x_t)}{p_{t,y}(x_t)}$ and $p_{s|t,y}(x_s|x_t) = \frac{p_{(s,t),y}(x_s,x_t)}{p_{t,y}(x_t)}$ we get that

$$\nabla \log p_{t,y}(x_t) = e^{(t-s)/2} \int_{\mathbb{R}^D} p_{s|t,y}(x_s|x_t) \nabla \log p_{s,y}(x_s) dx_s$$
$$= e^{(t-s)/2} \mathbb{E}_y \big[ \nabla \log p_{s,y}(X_s) | X_t = x_t \big].$$

$\square$

## C.3 Proof of Proposition 2

The proof adapts the one of [24] to the conditional setting. The only difference is that the expectation is $\mathbb{E}_y$, which affects the distribution of $X_0$ but not the one of $W$. Moreover, Lemma 4 shows that the key to the proof (the reverse-time martingale property of the finite-dimensional score) is still valid. Here $\mathbb{E}_y$ is the expectation with respect to the distribution of $X_0$ and $W$ given $Y = y$.

To prove Proposition 2, we are left to show that the solution of the reverse-time SDE

$$dZ_t = \frac{1}{2}Z_t dt + S(T - t, Z_t, y)dt + \sqrt{C}dW_t, \quad Z_0 \sim X_T|Y = y \tag{34}$$

satisfies $Z_T \sim X_0|Y = y$. We recall that $X_t$ is the solution to the SDE

$$dX_t = -\frac{1}{2}X_t dt + \sqrt{C}dW_t, \quad X_0 \sim \mu_0.$$

We first notice that $X_t$ is given by the following stochastic convolution:

$$X_t = e^{-t/2}X_0 + \int_0^t e^{-(t-s)/2}\sqrt{C}dW_s.$$

For $P^D$ the orthogonal projection on the subspace of $H$ spanned by $v_1, \ldots, v_D$ (the eigenfunctions of $C$), $X_t^D = P^D(X_t)$ are solutions to

$$dX_t^D = -\frac{1}{2}X_t^D dt + \sqrt{(C^D)}dW_t^D,$$

where

$$C^D = P^D C P^D, \quad W_t^D = P^D W_t.$$

We define $X_t^{D:M} = X_t^M - X_t^D$. Then

$$X_t^{D:M} = e^{-t/2}X_0^{D:M} + \int_0^t e^{-(t-s)/2}\sqrt{(C^{D:M})}dW_s^{D:M},$$

where the superscript $D : M$ indicates the projection onto $\mathrm{span}\{v_{D+1}, \ldots, v_M\}$. It holds that

$$\mathbb{E}_y\Big[\sup_{t \leq T}\|X_t^{D:M}\|_H^2\Big] \leq 2e^{-t}\mathbb{E}_y[\|X_0^{D:\infty}\|_H^2] + 2(1 - e^{-t})\sum_{i=D+1}^{\infty}\lambda_i \to 0$$

as $D \to \infty$, where we used Doob's $L^2$ inequality to bound the stochastic integral. Therefore $(X_t^N)$ is a Cauchy sequence and converges to $X_t$ in $L^2(\mathbb{P}_y)$. Consequently, the distribution of $X_t^N$ given $Y = y$ converges to the distribution of $X_t$ given $Y = y$ as $N \to +\infty$.

Recall that

$$S(t, X_t, y) = -(1 - e^{-t})^{-1}\mathbb{E}_y[X_t - e^{-t/2}X_0 \mid X_t],$$

and recall that

$$C^D\nabla \log p_{t,y}^D(X_t^D) = -(1 - e^{-t})^{-1}P^D\mathbb{E}_y[X_t - e^{-t/2}X_0 \mid X_t^D].$$

In particular, due to the tower property of the conditional expectations,

$$C^D\nabla \log p_{t,y}^D(X_t^D) = \mathbb{E}_y[S(t, X_t, y) \mid X_t^D].$$

Since, by Assumption 1,

$$\mathbb{E}_y[\|S(t, X_t, y)\|_H^2] < \infty,$$

the quantities $\mathbb{E}_y[X_t - e^{-t/2}X_0 \mid X_t^D]$ are bounded in $L^2(\mathbb{P}_y)$ and will converge to the limit, $\mathbb{E}_y[S(t, X_t, y) \mid X_t] = S(t, X_t, y)$, by the Martingale convergence theorem. We get rid of the projection $P^D$ by

$$(1 - e^{-t})\mathbb{E}_y[\|C^D\nabla \log p_{t,y}^D(X_t^D) - S(t, X_t, y)\|_H^2]$$

$$= \mathbb{E}_y[\|P^D\mathbb{E}_y[X_t - e^{-t/2}X_0 \mid X_t^D] - \mathbb{E}_y[X_t - e^{-t/2}X_0 \mid X_t]\|_H^2]$$

$$\leq \mathbb{E}_y[\|\mathbb{E}_y[X_t - e^{-t/2}X_0 \mid X_t^D] - \mathbb{E}_y[X_t - e^{-t/2}X_0 \mid X_t]\|_H^2]$$

$$+ \mathbb{E}_y[\|(I - P^D)\mathbb{E}_y[X_t - e^{-t/2}X_0 \mid X_t^D]\|_H^2]$$

$$\leq \mathbb{E}_y[\|\mathbb{E}[X_t - e^{-t/2}X_0 \mid X_t^D] - \mathbb{E}_y[X_t - e^{-t/2}X_0 \mid X_t]\|_H^2]$$

$$+ \mathbb{E}_y[\|(I - P^D)(X_t - e^{-t/2}X_0)\|_H^2].$$

The first term vanishes due to our previous discussion. The second term vanishes since

$$\mathbb{E}_y[\|(I - P^D)(X_t - e^{-t/2}X_0)\|_H^2] = \mathbb{E}_y[\|(I - P^D)\int_0^t e^{-(t-s)/2}\sqrt{C}dW_s\|_H^2]$$

$$= \mathbb{E}[\|(I - P^D)\int_0^t e^{-(t-s)/2}\sqrt{C}dW_s\|_H^2]$$

$$= (1 - e^{-t})\sum_{i=D+1}^{\infty}\lambda_j \to 0$$

as $D \to \infty$.

We now make use of the fact that $\nabla \log p_{t,y}^D$ is a square-integrable Martingale in the reverse-time direction by Lemma 4. We therefore get a sequence of continuous $L^2$-bounded Martingales converging to a stochastic process. Since the space of continuous $L^2$-bounded martingale is closed and pointwise convergence translates to uniform convergence, we get that $S$ is a $L^2$-bounded martingale, with the convergence of of $C^D \nabla \log p_{t,y}^D$ to $S$ being uniform in time.

We have that

$$Z_t^D - Z_0^D - \frac{1}{2}\int_0^t Z_s ds - \int_0^t C^D \nabla \log p_{s,y}^D(Z_s) = \sqrt{(C^D)}W_t^D.$$

Since all the terms on the left-hand side converge in $L^2$, uniformly in $t$, so does the right-hand side. Using again the closedness of the spaces of Martingales and Levy's characterization of Wiener process, we find that $\sqrt{(C^D)}W_t^D$ converges to $\sqrt{C}W_t$. Therefore

$$Z_t = Z_0 + \frac{1}{2}\int_0^t Z_s ds + \int_0^t S(t, Z_t, y) + \sqrt{C}W_t.$$

Therefore, $Z_t$ is indeed a solution to (34) and $Z_T \sim X_0|Y = y$. Using uniqueness of the solution we then conclude that this holds for any solution $Z_t$.

### C.4  Proof of (23)

$$\mathbb{E}[\|S(t, X_t, y)\|_H^2|Y = y] = (1 - e^{-t})^{-2}\mathbb{E}[\|\mathbb{E}[X_t - e^{-t/2}X_0|Y = y, X_t]\|_H^2|Y = y]$$

$$= (1 - e^{-t})^{-2}\mathbb{E}[\|\mathbb{E}[\int_0^t e^{-(t-s)/2}\sqrt{C}dW_s|Y = y, X_t]\|_H^2|Y = y]$$

$$\leq (1 - e^{-t})^{-2}\mathbb{E}[\mathbb{E}[\|\int_0^t e^{-(t-s)/2}\sqrt{C}dW_s\|_H^2|Y = y, X_t]|Y = y]$$

$$= (1 - e^{-t})^{-2}\mathbb{E}[\|\int_0^t e^{-(t-s)/2}\sqrt{C}dW_s\|_H^2|Y = y]$$

$$= (1 - e^{-t})^{-2}\mathbb{E}[\|\int_0^t e^{-(t-s)/2}\sqrt{C}dW_s\|_H^2]$$

$$= (1 - e^{-t})^{-1}\mathrm{Tr}(C).$$

### C.5  Proof of Proposition 3

Note that with the assumptions in Proposition 3 with $C_\mu$ and $C$ having the same basis of eigenfunctions and the separability assumption on the Radon-Nikodym derivative for the modes, the system for the modes again diagonalizes. However, in this case the (conditional) distribution for $X_0^{(j)}$ is non-Gaussian in general and the change of measure with respect to the Gaussian measure characterized by $\psi^{(j)}(x^{(j)}, y)$. We let the superscript $g$ denote the Gaussian case with $\psi \equiv 1$, then

we have:

$$S^{(j)}(t,x,y)$$

$$= -(1-e^{-t})^{-1}\left(x^{(j)} - e^{-t/2}\mathbb{E}[X_0^{(j)}|X_t = x, Y = y]\right)$$

$$= -(1-e^{-t})^{-1}\left(x^{(j)} - e^{-t/2}\int_\mathbb{R} x_0 \left(\frac{\mu_{x_0,x_t^{(j)}|y}^{(j,g)}(x_0,x^{(j)})\psi^{(j)}(x_0,y)}{\mu_{x_t^{(j)}|y}^{(j)}(x^{(j)})}\right) dx_0\right)$$

$$= (1-e^{-t})^{-1}\left(e^{-t/2}\int_\mathbb{R} x_0 \left(\frac{\mu_{x_0,x_t^{(j)}|y}^{(j,g)}(x_0+x^{(j)}e^{t/2},x^{(j)})\psi^{(j)}(x_0+x^{(j)}e^{t/2},y)}{\mu_{x_t^{(j)}|y}^{(j)}(x^{(j)})}\right) dx_0\right)$$

$$= S^{(j,g)}(t,x,y)T^{(j)}(t,x^{(j)},y) + \tilde{R}^{(j)}(t,x^{(j)},y),$$

for $S^{(j,g)}$ the mode score in the Gaussian case given in (31) and with

$$T^{(j)}(t,x^{(j)},y) = \left(\frac{\mu_{x_t^{(j)}|y}^{(j,g)}(x^{(j)})\psi^{(j)}(x^{(j)}e^{t/2},y)}{\mu_{x_t^{(j)}|y}^{(j)}(x^{(j)})}\right),$$

$$\tilde{R}^{(j)}(t,x^{(j)},y) = (1-e^{-t})^{-1}e^{-t/2}$$

$$\times \int_\mathbb{R} x_0 \left(\frac{\mu_{x_0,x_t^{(j)}|y}^{(j,g)}(x_0+x^{(j)}e^{t/2},x^{(j)})\left(\psi^{(j)}(x_0+x^{(j)}e^{t/2},y)-\psi^{(j)}(x^{(j)}e^{t/2},y)\right)}{\mu_{x_t^{(j)}|y}(x^{(j)})}\right) dx_0.$$

We have for $\phi_\lambda$ the centered Gaussian density at second moment $\lambda$

$$T^{(j)}(t,x^{(j)},y) = \frac{\int_\mathbb{R}\phi_{\lambda_t^{(j)}}(x^{(j)}-ve^{-t/2})\phi_{\mu_y^{(j)}}(v-x_y^{(j)})dv}{\int_\mathbb{R}\phi_{\lambda_t^{(j)}}(x^{(j)}-ve^{-t/2})\phi_{\mu_y^{(j)}}(v-x_y^{(j)})\psi^{(j)}(v,y)/\psi^{(j)}(x^{(j)}e^{t/2},y)dv},$$

for $\lambda_t^{(j)} = \lambda_j(1-e^{-t}), \mu_y^{(j)} = \mu_j/(1+q^{(j)})$ and $x_y^{(j)} = y^{(j)}q^{(j)}/(1+q^{(j)})$ for $y^{(j)} = \langle y, v_j\rangle$. Here $x_y^{(j)}, \mu_y^{(j)}$ are respectively the mean and variance of $X_0^{(j)}$ given $y$ and where we used the parameterization set forth in Section B.1. We then have

$$|T^{(j)}(t,x^{(j)},y)| \leq K^2, \quad \lim_{t\downarrow 0}T^{(j)}(t,x^{(j)}y) = 1.$$

We moreover have

$$|\tilde{R}^{(j)}(t,x^{(j)},y)| \leq e^{-t/2}(1-e^{-t})^{-1}L\left(\frac{\int_\mathbb{R}x_0^2\phi_{\lambda_t^{(j)}}(x^{(j)}-x_0e^{-t/2})\phi_{\mu_y^{(j)}}(x_0-x_y^{(j)})dx_0}{\int_\mathbb{R}\phi_{\lambda_t^{(j)}}(x^{(j)}-ve^{-t/2})\phi_{\mu_y^{(j)}}(v-x_y^{(j)})\psi^{(j)}(v,y)dv}\right)$$

$$\leq e^{-t/2}(1-e^{-t})^{-1}LK\left(\frac{\int_\mathbb{R}x_0^2\phi_{\lambda_t^{(j)}}(x^{(j)}-x_0e^{-t/2})\phi_{\mu_y^{(j)}}(x_0-x_y^{(j)})dx_0}{\int_\mathbb{R}\phi_{\lambda_t^{(j)}}(x^{(j)}-ve^{-t/2})\phi_{\mu_y^{(j)}}(v-x_y^{(j)})dv}\right).$$

We then find

$$\limsup_{t\downarrow 0}|\tilde{R}^{(j)}(t,x^{(j)},y)| \leq \lambda^{(j)}LK,$$

and, moreover

$$|\tilde{R}^{(j)}(t,x^{(j)},y)| \leq$$

$$\lambda_j LK e^{t/2}\left(\frac{1}{1+(e^t-1)p^{(j)}(1+q^{(j)})} + \lambda_j\frac{(x_y^{(j)}-x^{(j)}e^{t/2})^2}{(\mu_y^{(j)})^2}\frac{(e^t-1)}{(1+(e^t-1)p^{(j)}(1+q^{(j)}))^2}\right).$$

Consider in the Gaussian case in (22) a mode so that $p^{(j)}(1+q^{(j)}) \uparrow \infty$ and $\lambda_j$ fixed, then the contribution of this mode to the score norm blows up in the small time limit. The situation with

$p^{(j)}(1 + q^{(j)}) \uparrow \infty$ would happen for instance in a limit of perfect mode observation so that $\sigma_B \downarrow 0$ and thus $q^{(j)} \uparrow \infty$. Indeed in the limit of small (conditional) target mode variabilty relative to the diffusion noise parameter the score drift becomes large for small time to drive the mode to the conditional target distribution. We here thus assume $p^{(j)}(1 + q^{(j)})$ is uniformly bounded with respect to mode ($j$ index), moreover, that $C$ is of trace class. We then find that Assumption 1 is satisfied with the following bound

$$\sup_{t \in [0,T]} \mathbb{E}\big[\|S(t, X_t, y)\|_H^2 | Y = y\big]$$

$$\leq 2\left(\sum_j \lambda_j e^T \left(K^4 p^{(j)}(1 + q^{(j)}) + 2\lambda_j e^T (LK)^2 \left(1 + 3\left(p^{(j)}(1 + q^{(j)})(e^T - 1)\right)^4\right)\right)\right).$$

We remark that in the case that we do not have a uniform bound on the $p^{(j)}(1 + q^{(j)})$'s it follows from (23) that the rate of divergence of the expected square norm of the score is at most $t^{-1}$ as $t \downarrow 0$ with $C$ of trace class.

## C.6 Proof of Proposition 4

We start from (30):

$$\mathbb{E}_{x_t, y \sim \mathcal{L}(X_t, Y)}\big[\|S(t, x_t, y) - s_\theta(t, x_t, y)\|_H^2\big] = \mathbb{E}_{x_t, y \sim \mathcal{L}(X_t, Y)}\big[\|S(t, x_t, y)\|_H^2\big]$$
$$+ \mathbb{E}_{x_t, y \sim \mathcal{L}(X_t, Y)}\big[\|s_\theta(t, x_t, y)\|_H^2\big] - 2\mathbb{E}_{x_t, y \sim \mathcal{L}(X_t, Y)}\big[\langle S(t, x_t, y), s_\theta(t, x_t, y)\rangle\big].$$

From Definition 1 we have

$$\mathbb{E}_{x_t, y \sim \mathcal{L}(X_t, Y)}\big[\langle S(t, x_t, y), s_\theta(t, x_t, y)\rangle\big]$$
$$= -(1 - e^{-t})^{-1}\mathbb{E}_{x_t, y \sim \mathcal{L}(X_t, Y)}\left[\left\langle x_t - e^{-t/2}\mathbb{E}_{x_0 \sim \mathcal{L}(X_0|X_t = x_t, Y = y)}[x_0], s_\theta(t, x_t, y)\right\rangle\right]$$
$$= -(1 - e^{-t})^{-1}\mathbb{E}_{x_t, y \sim \mathcal{L}(X_t, Y)}\left[\mathbb{E}_{x_0 \sim \mathcal{L}(X_0|X_t = x_t, Y = y)}\left[\left\langle x_t - e^{-t/2}x_0, s_\theta(t, x_t, y)\right\rangle\right]\right]$$
$$= -(1 - e^{-t})^{-1}\mathbb{E}_{(x_0, x_t, y) \sim \mathcal{L}(X_0, X_t, Y)}\left[\left\langle x_t - e^{-t/2}x_0, s_\theta(t, x_t, y)\right\rangle\right].$$

We obtain that

$$\mathbb{E}_{x_t, y \sim \mathcal{L}(X_t, Y)}\big[\|S(t, x_t, y) - s_\theta(t, x_t, y)\|_H^2\big]$$
$$= B + \mathbb{E}_{(x_0, x_t, y) \sim \mathcal{L}(X_0, X_t, Y)}\big[\| -(1 - e^{-t})^{-1}(x_t - e^{-t/2}x_0) - s_\theta(t, x_t, y)\|_H^2\big],$$

with

$$B = \mathbb{E}_{x_t, y \sim \mathcal{L}(X_t, Y)}\big[\|S(t, x_t, y)\|_H^2\big] - \mathbb{E}_{(x_0, x_t) \sim \mathcal{L}(X_0, X_t)}\big[\|(1 - e^{-t})^{-1}(x_t - e^{-t/2}x_0)\|_H^2\big]$$

that does not depend on $\theta$. Since $\mathcal{L}(X_t|X_0 = x_0, Y = y) = \mathcal{L}(X_t|X_0 = x_0)$ we finally get that

$$\mathbb{E}_{x_t, y \sim \mathcal{L}(X_t, Y)}\big[\|S(t, x_t, y) - s_\theta(t, x_t, y)\|_H^2\big]$$
$$= B + \mathbb{E}_{x_0, y \sim \mathcal{L}(X_0, Y), x_t \sim \mathcal{L}(X_t|X_0 = x_0)}\big[\| -(1 - e^{-t})^{-1}(x_t - e^{-t/2}x_0) - s_\theta(t, x_t, y)\|_H^2\big].$$

# D  Numerical experiments: details and additional results

In this section, we provide additional details regarding our numerical experiments. In both experiments, we parameterize the conditional score $s_\theta(t, x_t, y)$ using discretization-invariant Fourier neural operators [FNOs; 53]. This parameterization enables mapping input triplets $(t, x_t, y)$ to the score conditioned on $y$ at time $t$. Once trained—by minimizing the objective function in equation (28) with respect to $\theta$—we use the FNO as an approximation to the conditional score to sample new realizations of the conditional distribution by simulating the reverse-time SDE in equation (14).

## D.1  Stylized example

In this example, the conditional distribution that we approximate is defined using the relation

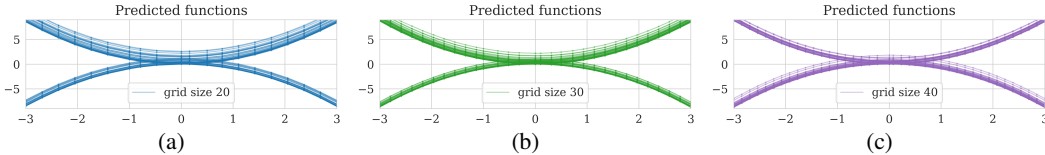

Figure 3: The proposed method's discretization invariance. Predicted samples (functions) on a uniformly sampled grid with (a) 20, (b) 30, and (c) 40 grid points.

$$x_0 = ay^2 + \varepsilon, \tag{35}$$

where $\varepsilon \sim \Gamma(1, 2)$ and $a \sim \mathcal{U}\{-1, 1\}$. Here, $\Gamma$ refers to the Gamma distribution, and $\mathcal{U}\{-1, 1\}$ denotes the uniform distribution over the set $\{-1, 1\}$. Having an explicit expression characterizing the conditional distribution allows us to easily evaluate the obtained result through our method—as opposed to needing to use a baseline method, e.g., Markov chain Monte Carlo.

**Training data** The discretization invariance of our model enables us to use training data that live on varying discretization grids. We exploit this property and simulate training joint samples $(x_0, y)$ by evaluating the expression in equation (35) over values of $y$ that are selected as nonuniform grids over the domain $[-3, 3]$ that contain 15–50 grid points.

**Architecture** In this example, the FNO comprises (i) a fully connected lifting layer that maps the three-dimensional vector—including the timestep $t$, the $y$ value, and the corresponding $x$ value—for each grid point to a 128-dimensional lifted space; (ii) five Fourier neural layers, as introduced in [53], which contain pointwise linear filters applied to the five lower Fourier modes; and (iii) two fully-connected layers, separated by a ReLU activation function, that map the 128-dimensional lifted space back to the conditional score (a scalar) for each grid point.

**Optimization details** To train the FNO, we minimized the objective function in equation (28) using $2 \times 10^4$ training steps. At each step, we simulated a batch of 512 training pairs selected from a grid with varying numbers of discretization points. We utilized the Adam stochastic optimization method [56] with an initial learning rate of $10^{-3}$, which decayed to $5 \times 10^{-4}$ during optimization, following a power-law rate of $-1/3$. Regarding the diffusion process, we followed the approach outlined in Ho et al. [29] and employed standard Gaussian noise with linearly increasing variance for the forward dynamics, which was discretized into 500 timesteps. The initial Gaussian noise variance was set to $10^{-4}$ and linearly increased over the timesteps until it reached $2 \times 10^{-2}$. The training hyperparameters were chosen by monitoring the validation loss over 1024 samples. The training process took approximately 7 minutes on a Tesla V100 GPU device. For further details, please refer to our open-source implementation on GitHub.

**Additional results** Figure 3 illustrates the predicted samples associated with the remaining testing grid sizes, whose densities were shown in Figures 1d–1f.

### D.2 Linearized seismic imaging example

The inverse problem we are addressing in this example involves the task of estimating the short-wavelength component of the Earth's unknown subsurface squared-slowness model using measurements taken at the surface. This particular problem, often referred to as seismic imaging, can be recast as a linear inverse problem when we linearize the nonlinear relationship between surface measurements and the squared-slowness model, as governed by the wave equation. In its simplest acoustic form, the linearization with respect to the slowness model—around a background smooth squared slowness model $m_0$—results in a linear inverse problem for the estimation of the true seismic image $\delta m^*$ using the following forward model,

$$d_i = J(m_0, q_i)\delta m^* + \epsilon_i, \quad \epsilon_i \sim p(\epsilon), \tag{36}$$

where $d = \{d_i\}_{i=1}^{n_s}$ represents a collection of $n_s$ linearized surface measurements, i.e., data wherein the zero-order term of Taylor's series has been subtracted, and $J(m_0, q_i)$ denotes the linearized

Born scattering operator, which is defined in terms of the source signature $q_i$ and the background squared-slowness model $m_0$. Due to noise and linearization errors, the above expression contains the term $\epsilon_i$. In addition to this noise term, the forward operator has a non-trivial nullspace due to the presence of shadow zones and finite-aperture data [57, 58]. To tackle this challenges, we set seismic imaging into a Bayesian framework and learn the associated posterior distribution via our proposed method.

**Training data**    We generated synthetic data by applying the Born scattering operator to 4750 2D seismic images, each with dimensions of $3075 \, \text{m} \times 5120 \, \text{m}$. These images were extracted from the Kirchhoff migrated Parihaka-3D dataset, which contains seismic images obtained by imaging the data collected in New Zealand [59, 60]. We parameterize the Born scattering operator using a background squared-slowness model (recall Figure 2b). The data acquisition geometry involves 102 sources with a spacing of $50 \, \text{m}$, each recorded for two seconds via 204 receivers spaced at $25 \, \text{m}$ located on top the image. The source wavelet used was a Ricker wavelet with a central frequency of $30 \, \text{Hz}$. To replicate a more realistic imaging scenario, we add band-limited noise to the the data, obtained by filtering white noise with the source function. To create training pairs, we first simulated noisy seismic data for all the 2D seismic images based on the aforementioned acquisition design. Subsequently, we reduce the dimensionality of seismic data by applying the adjoint of the Born scattering operator to the data. We use Devito [61, 62] for the wave-equation based simulations.

**Architecture**    In this example, the FNO is composed of (i) a fully connected lifting layer that maps the five-dimensional vector—including the timestep $t$, the two spatial coordinates, the data value ($y$, obtained after dimensionality reduction by applying the adjoint of the forward operator), and the corresponding $x$ value—for each grid point to a 32-dimensional lifted space; (ii) four Fourier neural layers, as introduced in [53], which contain pointwise linear filters applied to the 24 lower Fourier modes; and (iii) two fully-connected layers, separated by a ReLU activation function, that map the 128-dimensional lifted space back to the conditional score for each grid point.

**Optimization details**    We train the FNO according to the objective function in equation (28) with the Adam [56] stochastic optimization method with batch size 128 for 300 epochs. We use an initial stepsize of $2 \times 10^{-3}$, decaying to $5 \times 10^{-4}$ during optimization with a power-law rate of $-1/3$. We use a similar diffusion process as the previous example. The training hyperparameters are chosen by monitoring the validation loss over 530 samples. The training process takes approximately two hours and 15 minutes on a Tesla V100 GPU device. For further details refer to our open-source implementation on GitHub.

**Additional results**    Figures 4–7 illustrate more results regarding using the SDM to sample from the posterior distribution of multiple seismic imaging problem instances where the ground truth images (see Figures 4a–7a) are obtained from the test dataset. In each plot, we use the dimensionality-reduced data as the conditioning input to the FNO (see Figures 4b–7b). Through the SDM, we obtain $10^3$ posterior distribution samples and use them to estimate the conditional mean (see Figures 4c–7c) and the pointwise standard deviation among samples (see Figures 4d–7d), with the former serving as a measure of uncertainty. In all cases, the regions of significant uncertainty correspond well with challenging-to-image sections of the model, which qualitatively confirms the accuracy of our Bayesian inference method. This observation becomes more apparent in Figures 4g–7g, displaying two vertical profiles with 99% confidence intervals (depicted as orange-colored shading) for each experiment, which demonstrate the expected trend of increased uncertainty with depth. Furthermore, we notice that the ground truth (indicated by dashed black lines) mostly falls within the confidence intervals for most areas. We also consistently observe a strong correlation between the pointwise standard deviation and the error in the conditional mean estimate (see Figures 4e–7e), which further asserts the accuracy of our method in characterizing the posterior distribution in this large-scale Bayesian inference problem. To prevent bias from strong amplitudes in the estimated image, we present the normalized pointwise standard deviation divided by the envelope of the conditional mean in Figures 4f–7f. These visualizations provides an amplitude-independent assessment of uncertainty, highlighting regions of high uncertainty at the onset and offset of reflectors (both shallow and deeper sections). Additionally, the normalized pointwise standard deviation underscores uncertainty in areas of the image where there are discontinuities in the reflectors (indicated by black arrows), potentially indicating the presence of faults.

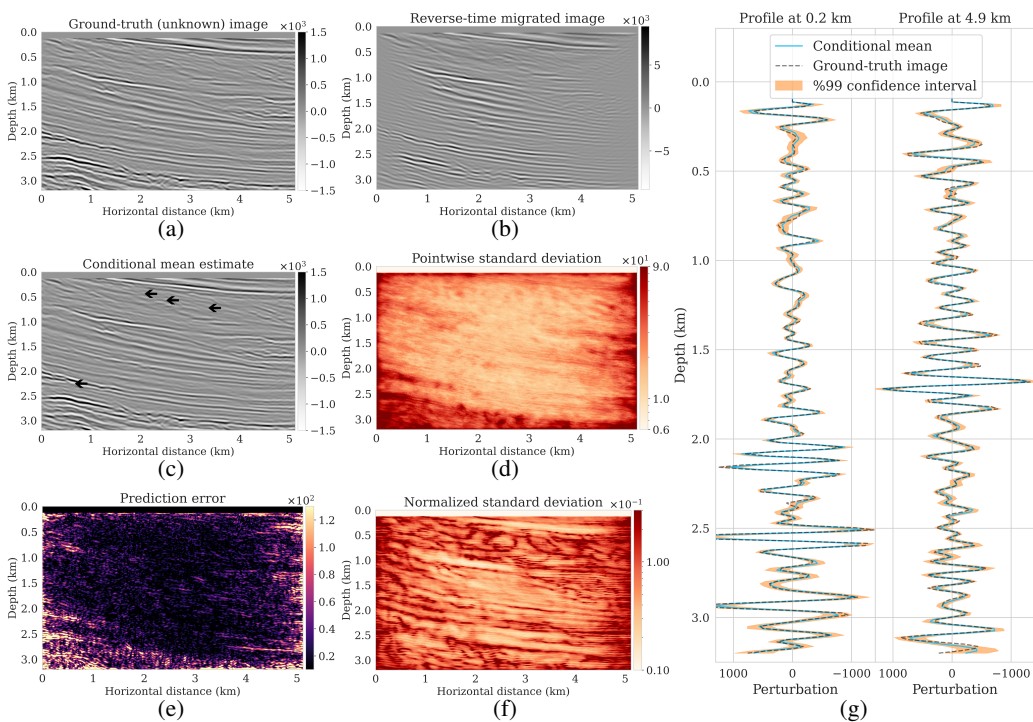

Figure 4: Seismic imaging and uncertainty quantification. (a) Ground-truth seismic image. (b) Data after applying the adjoint Born operator (known as the reverse-time migrated image). (c) Conditional (posterior) mean. (d) Pointwise standard deviation. (e) Absolute error between Figures 4a and 4c. (f) Normalized pointwise standard deviation by the envelope of the conditional mean. (g) Vertical profiles of the ground-truth image, conditional mean estimate, and the 99% confidence interval at two lateral positions in the image.

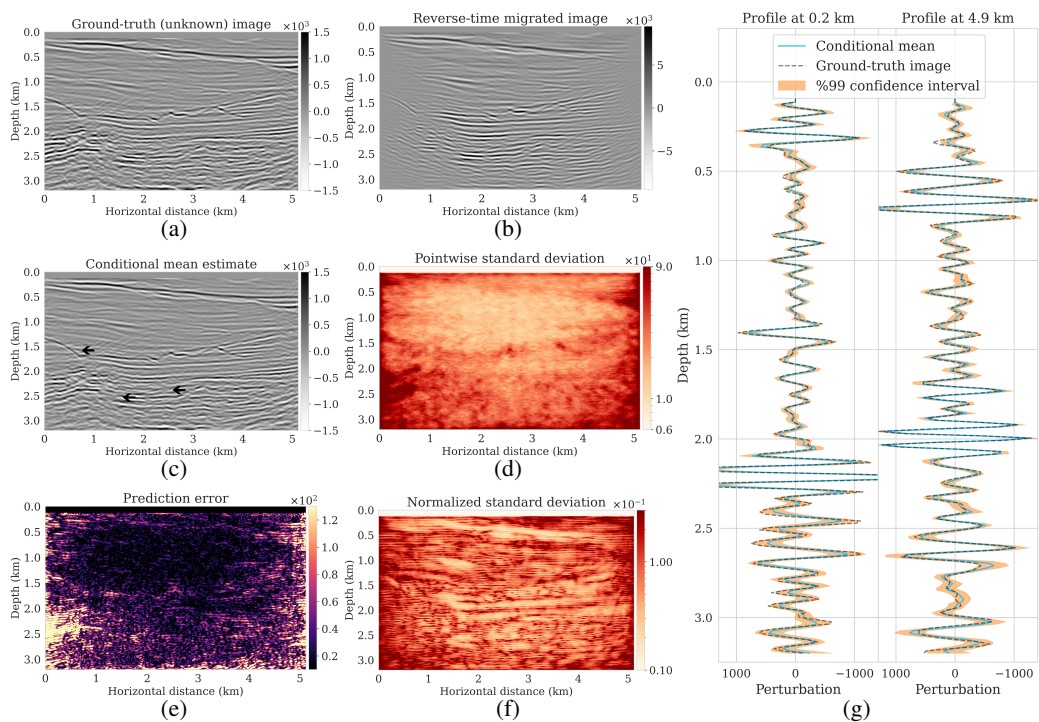

Figure 5: Seismic imaging and uncertainty quantification. (a) Ground-truth seismic image. (b) Data after applying the adjoint Born operator (known as the reverse-time migrated image). (c) Conditional (posterior) mean. (d) Pointwise standard deviation. (e) Absolute error between Figures 5a and 5c. (f) Normalized pointwise standard deviation by the envelope of the conditional mean. (g) Vertical profiles of the ground-truth image, conditional mean estimate, and the $99\%$ confidence interval at two lateral positions in the image.

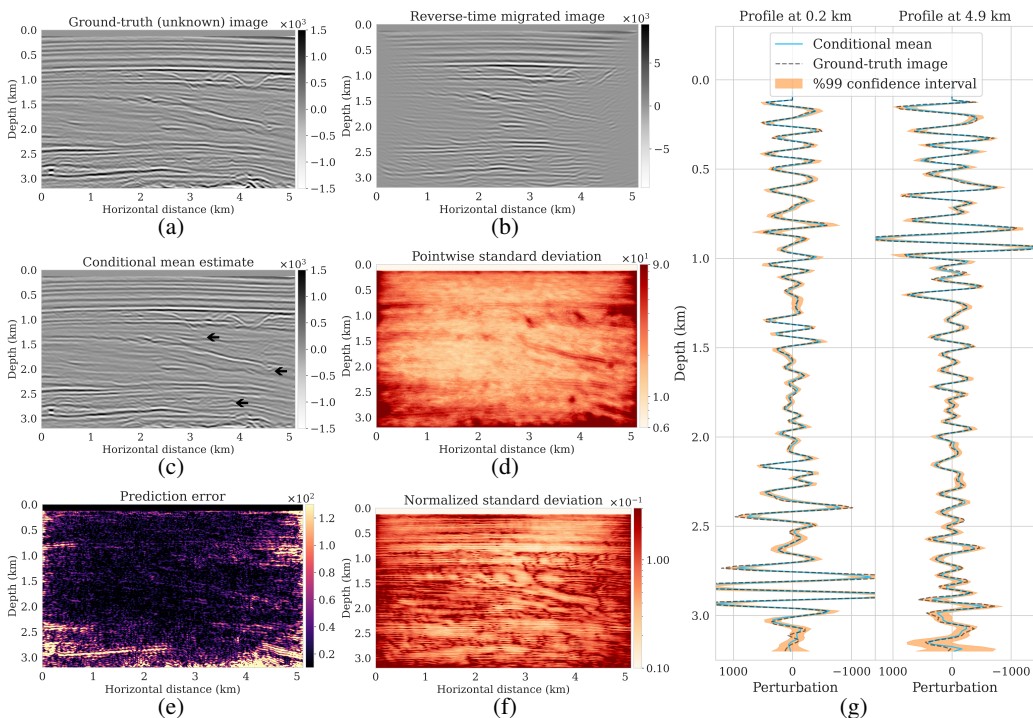

Figure 6: Seismic imaging and uncertainty quantification. (a) Ground-truth seismic image. (b) Data after applying the adjoint Born operator (known as the reverse-time migrated image). (c) Conditional (posterior) mean. (d) Pointwise standard deviation. (e) Absolute error between Figures 6a and 6c. (f) Normalized pointwise standard deviation by the envelope of the conditional mean. (g) Vertical profiles of the ground-truth image, conditional mean estimate, and the 99% confidence interval at two lateral positions in the image.

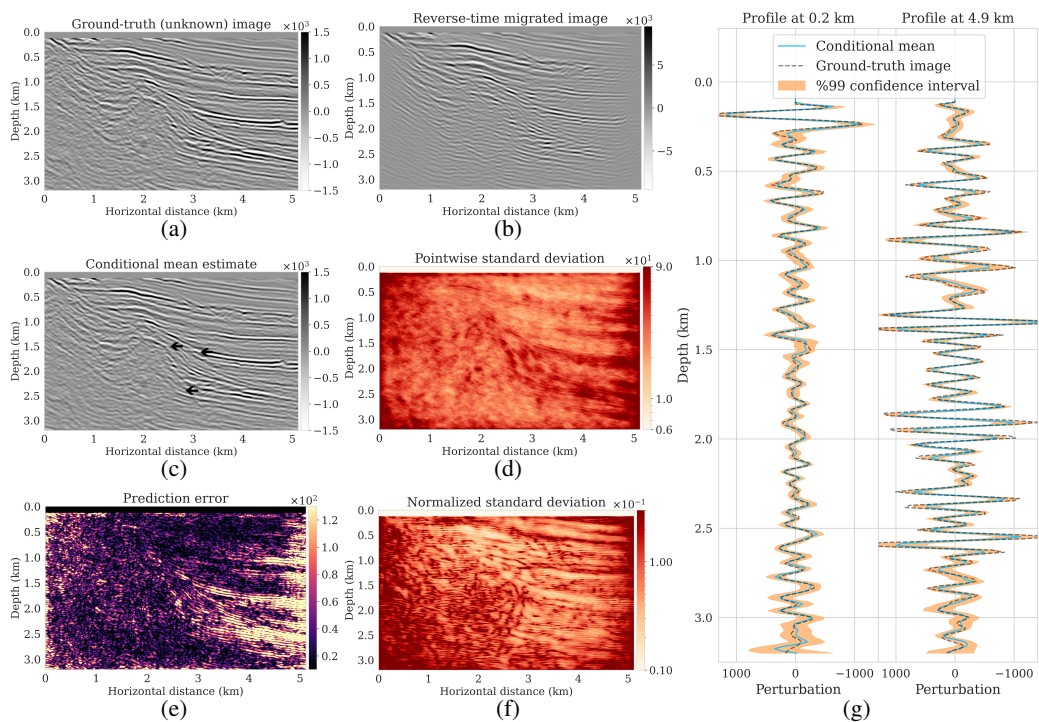

Figure 7: Seismic imaging and uncertainty quantification. (a) Ground-truth seismic image. (b) Data after applying the adjoint Born operator (known as the reverse-time migrated image). (c) Conditional (posterior) mean. (d) Pointwise standard deviation. (e) Absolute error between Figures 7a and 7c. (f) Normalized pointwise standard deviation by the envelope of the conditional mean. (g) Vertical profiles of the ground-truth image, conditional mean estimate, and the $99\%$ confidence interval at two lateral positions in the image.

