# OpenReview forum: "Conditional score-based diffusion models for Bayesian inference in infinite dimensions"
_NeurIPS.cc/2023/Conference — NeurIPS 2023 spotlight_

### Official Review · Reviewer_XSPZ · 2023-06-26

**Soundness:** 3 good
**Presentation:** 3 good
**Contribution:** 3 good
**Rating:** 7
**Confidence:** 2

**Summary:**

- The study proposes a method to learn the posterior distribution in infinite-dimensional Bayesian linear inverse problems using amortized conditional Score-based Diffusion Models (SDMs). This extends conditional SMDs into the infinite-dimensional function space setting, as existing conditional SDMs have previously only dealt with finite-dimensional vector spaces (noting also that _unconditional_ SDMs have recently been extended to infinite dimensional vector spaces by Pidstrigach et. al).  This leads the way for applications in, for example PDEs, where the unknown parameters to be estimated take the form of functions.
- The key technique underlying their approach is to define the _conditional_ score in an infinite dimensional setting, extending the method of Pidstrigach et. al, who defined the _unconditional_ score in the infinite dimensional setting.
- Using their definition of the conditional score in infinite dimensions, this allows them to avoid having to solve a potentially expensive proximal optimization step, as was done by Pidstrigach et. al.
- The authors then provide a comprehensive theoretical analysis of the use of their conditional score in SDMs and show:
  - How this newly defined score is used as a reverse drift of the diffusion process, which leads to a generative model that samples from the correct target conditional distribution under certain conditions.
  - That as long as you start from the invariant distribution of the diffusion process, the reverse SDE converges to the target distribution exponentially fast
  - By explicitly computing the expected square norm of the conditional score, they shows that a uniform in time estimate is not always true for the conditional score. This leads them to provide a set of conditions to be satisfied to ensure a uniform in time estimate for a general class of prior measures.
  - That the conditional score can be estimated via a conditional denoising score matching objective in infinite dimensions.
  - That the conditional denoising estimator is a consistent estimator of the conditional score in infinite dimensions.
  - That unlike the unconditional score, for noiseless observations the conditional score blows up as T->0
- They present a small toy experiment that validates their approach by demonstrating the applicability of their method in approximating non-Gaussian multi-modal distributions.





**Strengths:**

I will address strengths and weaknesses across the four dimensions (Originality, Quality, Clarity, Significance) below.

**Weaknesses:**

**Originality:**
- Are the tasks or methods new?
  - Yes, the authors make a novel contribution to the quickly-growing diffusion model literature, by showing how to extend conditional SDMs into the infinite-dimensional vector space, thus opening the door to a wider range of applications (PDEs, etc)
- Is the work a novel combination of well-known techniques? Is it clear how this work differs from previous contributions?
  - The authors are very clear about:
     1. How they are starting with the recently-proposed framework of Pidstrigach et al
     2. The specific point where they deviate from, and then extend, Pidstrigach's work (Definition 2, eq. 8)
  - Further, they provide a comprehensive analysis of the use of their defn 2 in SDMs in sections 4 and 5.
  - However I should state that I am not adequately familiar with the mathematical techniques deployed in this paper to check their technical claims for correctness or novelty, so please differ to another reviewer with more expertise in this area.
- Is related work adequately cited?
  - Yes.

**Quality:**
- Is the submission technically sound?
  - It appears to be, although as previously stated I am not sufficiently versed in their mathematical techniques to be 100% sure.
- Are claims well supported (e.g., by theoretical analysis or experimental results)?
  - Theoretically yes, experimentally no. I realize this is a theory paper and don't necessarily expect full-scale experiments on PDEs, but in Section 6 I expected to see at least 2 obvious baselines which were foreshadowed in the body of the text but not experimentally validated on the toy experiments. These are:
    1. How does their method compare to the crude, discretization-based approach discussed in lines 56 - 61:
        > A straightforward solution may be to discretize the infinite-dimensional input and output function spaces into finite-dimensional vectors, and apply SDMs to learn the posterior. Yet theoretical studies of current DMs suggest that performance guarantees do not generalize well on increasing dimensions [7, 9, 33]. This is precisely why Stuart’s guiding principle to study a Bayesian inverse problem for functions— “avoid discretization until the last possible moment” [41] — is more than ever critical to the use of SDMs.
    A primary motivation for their method is "Stuart's Principle", which says to avoid discretization until the last possible moment. I would have liked to see experimentally why this principle is so important on the simple toy examples they have provided.
    2. How does their method compare to the approach proposed by Pidstrigach for conditional sampling, which uses a proximal optimization step? This is a second primary motivation for their approach (and appears in the abstract) - their method is more performant because it avoids solving an optimization problem at each timestep. As far as I can tell it has not been experimentally validated that their approach is faster or more performant than the baseline method of Pidstrigach. Evidence showing their approach either gets better samples, or get samples of the same quality but more efficiently, is needed, given that a primary motivation for their approach is that the baseline method of Pidstrigach may be too computationally costly because of their use of proximal optimization.
- Are the methods used appropriate?
  - Yes
- Is this a complete piece of work or work in progress?
  - Yes, modulo the missing baselines discussed above. I believe such baselines are needed to consider this a complete piece of work, given how "Stuart's Principle" and "avoid proximal optimization" play a key role in the storyline and motivation for their technique.
- Are the authors careful and honest about evaluating both the strengths and weaknesses of their work?
  - Yes. In particular I found their observation, that unlike the unconditional score, for noiseless observations the conditional score blows up as t->0, particularly interesting. However in the conclusion I would have liked to read more about the authors' reflections on the strengths/weaknessness/future directions of their approach, both from a technical standpoint and from the standpoint of potential downstream applications of this work.

**Clarity:**
- Is the submission clearly written?
  - I found all writing up until section 4 relatively easy to follow. I began to get lost around sections 4 and 5 and couldn't follow the math, but I attribute this largely to not being comfortable with stochastic differential equations. I could still follow the high-level plot in these sections, but will defer to other reviews to evaluate the technical claims.
- Is it well organized?
  - Yes
- Does it adequately inform the reader?
  - Yes

**Significance:**
- Are the results important?
  - Yes, although I would have liked the authors to motivate the applications of their approach more thoroughly. I believe they only listed PDEs as an example for why you would want to use this approach but surely there are more applications than just PDEs, no?
- Are others (researchers or practitioners) likely to use the ideas or build on them?
  - Yes
- Does the submission address a difficult task in a better way than previous work?
  - Theoretically it appears so, but baselines are needed to experimentally validate these claims.
- Does it advance the state of the art in a demonstrable way?
  - Theoretically it appears so, but baselines are needed to experimentally validate these claims.
-  Does it provide unique data, unique conclusions about existing data, or a unique theoretical or experimental approach?
  - Yes, they build on the work of Pidstrigach in a novel way.


**Questions:**

- What are some other applications of this work besides PDEs? Can you include a few of them in the introduction so the reader doesn't think applications of your work is limited only to PDEs?

- In the conclusion the authors say their method "is able to perform conditional sampling directly on infinite dimensional Hilbert spaces." Don't Gaussian processes also allow you to perform conditional sampling directly on infinite dimensional Hilbert spaces? What are the differences between your approach and Gaussian processes? When would I want to use one vs. the other? Could a GP be used as a baseline in your Figure 1? If so it would be very interesting to see how it compares.

- The authors also say "We also show that the conditional score can have a singular behavior at small times when the observations are noiseless, in contrast with the unconditional score under similar hypotheses.". Is this just a limiting phenomenon of theoretical interest, or do you expect this to cause difficulties in practice? Are there any applications where we should expect observations to be noiseless or does that assumption never hold in practice?


**Limitations:**

- There does not appear to be a "Limitations and Broader Impacts" statement in this work.

---

> ### Author Rebuttal · Authors · 2023-08-09
>
> We would like to thank the Reviewer for their feedback.
>
> We are happy to clarify our manuscript in response to the Reviewer's questions. We hope that this could lead to an improvement in their assessment of the paper.
>
> 1. **Discretization-based approaches and Pidstrigach's procedure.** While it would have been interesting to extend the section dedicated to numerical experiments with explicit comparison with other methods, we want to point out that the limitations of crude discretization-based approaches are well-documented in the literature on Bayesian inverse problems. Consequently, we made a deliberate decision to avoid focusing on them experimentally. Moreover, in the literature on score-based diffusion models (SDMs), there are already instances where results show degradation when generalizing to infinity [1, Figure 5].
>
>     Regarding the comparison between our method and the one by Pidstrigach [2], we want to stress that our method is not necessarily more performant$-$that was never our claim. It really depends on the task to be solved. Our aim was to frame the comparison between the two methods through the lens of the more general comparison between case-specific inference and amortized inference. Amortized methods *can* be a preferred option in Bayesian inverse problems. In our paper, we cite some works where these examples are provided (line 119).
>
>    Given that there is abundant literature on both topics---the importance of discretizing as late as possible and amortized methods---we believe that it is more interesting, since this is a theory paper, to focus on the fact that while Pidstrigach's method for conditional sampling is heuristic, ours *provably* samples from the posterior and is discretization-invariant. Our main contribution is to offer theoretical guarantees for sampling from the posterior, with the additional appeal of providing a method for practitioners seeking discretization-invariant amortized DMs. In this sense, it is important to note that our method stands in comparison to Pidstrigach's one. In fact, their implementation is essentially finite. Pidstrigach use a UNet to parametrize their score [2, Section 6], which restricts its evaluation to the training interval. If dealing with a specific inverse problem on a new grid, they would need to gather new training data, retrain the NN, and limit the use of the score to that specific grid. In contrast, our approach allows us to move away from the initial grid, effectively taking advantage of the discretization-invariance property. This flexibility ensures a broader applicability of our method. Your comment helped us recognizing that we haven't stressed this difference enough, so we will add a remark and a new numerical experiment (see Figure 1 in the PDF included in our global response).
>
>    Finally, we agree with your remark that our toy example needs some improvement to better fit the storyline. In the PDF we included preliminary results, namely Figure 1, illustrating that the method can handle different discretizations. This serves to underscore the importance of Stuart’s principle. Additionally, we incorporated an experiment on a large-scale inverse problem in geophysics, specifically linearized seismic imaging via the Born approximation with a $256 \times 256$-dimensional unknown parameter, to enhance the comprehensiveness of our results (see Figure 2).  We will add these experiments in the paper.
>
> 2. **Concluding remarks.** Given the theoretical nature of the paper, summarizing a series of details from the proofs and presenting them in a self-contained section is challenging. Nonetheless, we recognize the value in discussing the strengths, weaknesses, and future directions of our method. Therefore, we intend to address this aspect by adding a few remarks throughout the text incorporating, among others, Reviewer hg8t's second question.
>
> 3. **Applications beyond PDEs.** While applications of our work to PDE-based inverse problems are the most natural ones, they are by no means the only ones. In the non-PDE class we may think at geometric inverse problems (e.g. how to determine the Riemann metric from geodesic information, or the background velocity map from travel time information in geophysics) [3] or inverse problems for singular integral operators [4]. We will cite these examples in our paper.
>
> 4. **Can a GP be used as a baseline in Figure 1?** In the Gaussian framework of Section 4 this would make sense but in the general framework of Sections 5-6 this does not seem possible. It is clear that the conditional distribution in the example of Section 6 is strongly bimodal so GP regression does not seem appropriate to address this example.
>
> 5. **Singularity of the conditional score.** The singularity in the conditional score as noise vanishes has been investigated in finite dimensions. It is not merely a theoretical phenomenon but has practical implications. For a discussion about the difficulties this singularity causes and numerical methods for approximating the score at small times in finite dimensions, we refer to [5, 6]. As for our paper, we anticipate that Assumption 1 is reasonably easy to satisfy. However, we must be mindful of the possibility of encountering a blow-up of the score under the conditions described by Reviewer hg8t. To address this concern, we will add a remark.
>
> **References**
>
> [1] A. Phillips et al, *Spectral diffusion processes*, 2022.
>
> [2] J. Pidstrigach et al, *Infinite-dimensional diffusion models for function spaces*, 2023.
>
> [3] G. Uhlmann and A. Vasy, *The inverse problem for the local geodesic ray transform*, 2016.
>
> [4] A. Dynin, *Inversion problem for singular integral operators:* $C^*$*-approach*, 1978.
>
> [5] D. Kim et al, *Soft Truncation: A Universal Training Technique of Score-based Diffusion Model for High Precision Score Estimation*, 2022.
>
> [6] T. Dockhorn et al, *Score-Based Generative Modeling with Critically-Damped Langevin Diffusion*, 2022.

---

> > ### Comment · Reviewer_XSPZ · 2023-08-11
> > **Thank you for your response**
> >
> > Thank you for your response, which I found convincing. Given the new (and impressive!) experiments, I've happily increased my score from 6->7.

---

### Official Review · Reviewer_Q2NB · 2023-06-29

**Soundness:** 3 good
**Presentation:** 3 good
**Contribution:** 3 good
**Rating:** 8
**Confidence:** 3

**Summary:**

This paper mathematically examines linear inverse problems in infinite dimensional vector spaces. Particularly, it is proved that the conditional denoising estimator is a consistent estimator of the conditional score in infinite dimension.

**Strengths:**

The consistency of the conditional denoising estimator in infinite dimensional vector spaces is mathematically shown. For a specific case of Gaussian prior, the forward-reverse SDEs are solved exactly, which shows an exponentially fast convergence in the reverse SDE. A sufficient condition for the success of the score-based diffusion model framework is presented for the infinite dimensional version.

**Weaknesses:**

The considered inverse problems are not of the plug-and-play type, which would limits the practical utility. Numerical example is limited to a one-dimensional toy model.


**Questions:**

Gaussian process (GP) is a popular method for inverse problem in Hilbert space. I wonder if the current problem would have a certain connection to GP, in particular, in the case of Gaussian prior discussed in section 4.

**Limitations:**

Descriptions on the motivation to examine the inverse problem in Hilbert space are lacking.  In what practical situations do inverse problems in Hilbert space come out? For instance, GP is widely used for Bayes optimization, which is an optimization scheme for black-box functions. Additional writing about possible applications of the inverse problem in Hilbert space would make the paper more attractive.

---

> ### Author Rebuttal · Authors · 2023-08-09
>
> We would like to thank the Reviewer for their positive feedback.
>
> We are happy to clarify our manuscript in response to the Reviewer's remarks and questions.
>
> 1. **Practical utility against plug-and-play-type approach.** While plug-and-play methods are indeed very popular, we would like to emphasize that we deliberately chose not to focus on them, first and foremost because we wanted to provide theoretically-grounded guarantees for an approach that is not heuristic. Furthermore, it is worth noticing that current implementation of plug-and-play approaches using diffusion models for conditional sampling in the infinite-dimensional setting [1, Section 6] does not fully exploit the ``discretization-invariance'' property achieved by studying the problem in infinite dimensions. Pidstrigach employ a UNet to parametrize their score, which restricts the evaluation of their score function to the training interval. Consequently, when dealing with a specific inverse problem on a new grid, they would need to gather new training data on that grid, retrain the neural network, and consequently limit the use of the score to that specific grid. In contrast, our implementation allows us to handle new discretizations without requiring additional training data, as demonstrated in the new experiment that we intend to include in the final version of the paper (see Figure 1 in the attached PDF for preliminary results). Therefore, we believe that our approach is not only theoretically grounded but also offers a broader applicability for our method.
>
> 2. **Numerical example is limited to a one-dimensional toy model.** We acknowledge your remark, along with those of the other reviewers, regarding the need for
> improvement in our toy example. In our global response, we have included a PDF that
> contains additional experiments, illustrating the applicability of our method to a large-scale inverse problem in geophysics, specifically linearized sesmic imaging via the Born
> approximation (Figure 2 in the attached PDF) with a $256 \times 256$-dimensional unknown parameter. We will add these experiments in the paper.
>
> 3. **Connection to Gaussian Process.** Gaussian process regression is indeed related to the calculations presented in section 4 which is in the Gaussian framework. GP has been used to solve inverse problems because of - at least - two fundamental reasons. First it naturally arises when data is contaminated by noise, and additive Gaussian  noise is the most simple noise model. Second it allows the use of simple but powerful theorems (such as Gaussian conditioning theorem). Moreover, in the infinite-dimensional setting, the GP approach is possible via the use of Gaussian measures (and appropriate theorems such as Feldman-Hajek theorem). In our paper we leverage these aspects of Gaussian analysis to gain novel and fundamental insights about the conditional score. Moreover, we emphasize the robustness of these insights  by introducing a prior that is absolutely continuous with respect to Gaussian measure.
>
> 4. **Possible applications of inverse problems in Hilbert spaces.** Bayesian approach to inverse problems makes it possible to deal with under-determined and/or noisy inverse problems by an appropriate prior modelling. In the infinite-dimensional PDE context, this prior should sample functions in the suitable space functions, which are very often Hilbert spaces. For instance, the inverse heat equation (how to determine the initial condition of a heat or convection-diffusion equation from noisy measurements) or the elliptic inverse problem (how to determine the source of an elliptic equation from noisy measurements) presented in [2] are concrete examples of noisy and under-determined inverse problems and they are naturally formulated in Hilbert spaces. We will mention these examples in the paper.
>
> **References**
>
> [1] J. Pidstrigach, Y. Marzouk, S. Reich, and S. Wang, *Infinite-dimensional diffusion models for function spaces*, arXiv:2302.10130.
>
> [2] M. Dashti and A. M. Stuart, *The Bayesian approach to inverse problems*, in Handbook of Uncertainty Quantification, Springer, 2017, pp. 311-428.

---

> > ### Comment · Reviewer_Q2NB · 2023-08-17
> >
> > Thank you for the reply. I keep my evaluation as it is.

---

### Official Review · Reviewer_WxYS · 2023-07-05

**Soundness:** 3 good
**Presentation:** 3 good
**Contribution:** 2 fair
**Rating:** 6
**Confidence:** 3

**Summary:**

This paper proposed a method to deal with inverse problems in infinite dimensions using conditional-score-based models. Specifically, they propose to directly learn the posterior distribution in infinite-dimensional Bayesian linear inverse problems using amortized conditional SDMs. Moreover, this paper also discussed the robustness of the learned distribution against perturbations of the observations.  A numerical experiment is conducted to validate the efficiency.

**Strengths:**

1. This paper proposed an interesting method to deal with infinite-dimensional Bayesian linear inverse problems.

2. It provides a detailed analysis of the forward-reverse conditional SDE framework in the case of a Gaussian prior measure.

3. It provides a set of conditions to ensure a uniform in-time estimate for a general class of prior measures.

**Weaknesses:**

1. Regarding the introduced definition of the conditional score and the result that the conditional score can be estimated via a conditional denoising score matching, it seems that they are straightforward extensions of the unconditional case. For score-based models, the key is to learn a general distribution using score matching, regardless of whether it is a conditional distribution or a non-conditional distribution. I mean, there is actually no fundamental difference between a conditional distribution and non-conditional distribution, i.e., score-based matching applies to them or other general distributions equally.

2. From my understanding, and also as suggested in the abstract, this paper only focuses on infinite-dimensional Bayesian linear inverse problems, rather than arbitrary inverse problems. The title of this paper is kind of misleading.  On the other hand, for linear inverse problems, the score of the likelihood term is relatively easy to obtain, compared to directly training a conditional score network. Please correct me if I am wrong.

**Questions:**

Some additional questions:

1. What is the difference between (2) and (3)?

2. The experiment part only considers a simple low-dimensional problem, is it possible to add some results of large-scale problems?

**Limitations:**

See above.

---

> ### Author Rebuttal · Authors · 2023-08-09
>
> We thank the Reviewer for their feedback.
>
> We are happy to clarify our manuscript in response to the Reviewer's questions. We hope that this could lead to an increase in their score.
>
> 1. **Conditional SDMs.** Various approaches have been proposed in the literature for dealing with conditioning, both in finite and infinite-dimensional cases. In [1], an effective approach involves subspace projection during time-reversed diffusion. The subsequent noise-level-dependent update yields favorable application outcomes. In [2] three methods are discussed from theoretical and practical viewpoints in finite dimensions. It is indeed not obvious what is the best way to enforce the constraint associated with observations and what the role of the noise level is, and how to incorporate the constraints from a computational perspective. These challenges are particularly acute in the infinite-dimensional case (see next answers). Here we remark that we agree that a Bayesian and a conditional score perspective is a natural approach. It is important however to understand how the conditioning affects the score and the training and the role of noise. Here we have shed light on this important challenge by considering the Gaussian context discussed in Section 4 where we get explicit insight about the conditional score. This reveals in particular the singular behavior that one may observe for the time reversed diffusion in case of low noise observations. Moreover, we generalized the results in [2] regarding a conditional denoising estimator in infinite dimensions. This result is important for efficient training and provides a theoretical underpinning for efficient conditional generation and further research. Indeed in the context of inverse problems efficient implementation of the conditioning is the central challenge.
>
> 2. **Title.** Regarding the potentially misleading title, we don't know if it can be modified at later stages. While we agree that we could have been more vocal in the title about the specific problems we are addressing, we remark that the vast majority of papers concerning SDMs and inverse problems (IPs) focus on linear IPs. Addressing nonlinear IPs is a challenging task, and the limited subset of papers that delve into more general cases typically highlight this in their titles from the outset. However, we will make a concerted effort to emphasize in the abstract and throughout the text that we are exclusively considering linear IPs.
>
> 3. **Score of the likelihood.** Contrarily to methods that incorporate the gradient of the log-likelihood in order to sample from the posterior, ours does not assume the knowledge of the forward model, as only data pairs are used to learn the score. We remark that, while using the log-likelihood in SDMs linear IPs looks straightforward, it becomes in fact analytically intractable in terms of DMs, due to their dependence on time. That's why existing approaches resort to projections onto the measurement subspace [3, Section 5.4]. However, we remark that Pidstrigach's implementation [3, Section 6] utilizes UNet to parameterize the score, limiting the evaluation of the score function to the training interval only. In contrast, our method is not limited to the grid on which we initially train our network. In other words, we truly incorporate the advantages of the infinite-dimensional approach. Additionally, we remark that the projection-type methods are primarily heuristic and suffer from instability when dealing with ill-posed IPs. Recently, some workarounds have been proposed to tackle these issues in finite dimensions [4, 5], but we remark that our method, together with not assuming any knowledge of the forward model and operating directly in infinite dimensions, is also designed to offer an alternative to data-specific inference. In fact, projection-type methods can involve costly forward operator computations. Our method, instead, learns an amortized version of the conditional score and, by doing so, addresses a critical gap in the literature involving infinite-dimensional SDMs. There exist cases in which amortized methods can be a preferred option for Bayesian IPs (see line 119 and accompanying references in our paper).
>
> 3. **Difference between (2) and (3).** In finite dimensions, the drift of the reverse stochastic differential equation (SDE) involves the score function $\nabla \log p_t$, where $p_t$ represents a density with respect to a Lebesgue measure. However, in infinite-dimensional Hilbert spaces, this density is no longer well-defined (the Heine-Borel theorem does not hold in such spaces). As a result, the left-hand side of equation (2) cannot be interpreted literally in infinite dimensions. In equation (3), $S(t,x)$ is thus defined formally, leveraging the fact that the right-hand side of equation (2) is well-defined in infinite dimensions. This implies that we need to demonstrate that $S(t,x)$ truly represents the score function, i.e., the drift of the reverse SDE. Analyzing the forward-reverse conditional SDE framework is an important contribution of our paper.
>
> 4. **Is it possible to add results on large-scale problems?** Yes, see the PDF included in our global response. Figure 2 illustrates a new experiment, which we will include in the appendix, demonstrating the applicability of our method to a large-scale inverse problem in geophysics, i.e. linearized seismic imaging via the Born approximation with a $256 \times 256$-dimensional unknown parameter.
>
> **References**
>
> [1] Y. Song et al, *Solving inverse problems in medical imaging with score-based generative models*, 2022.
>
> [2] G. Batzolis et al, *Conditional image generation with score-based diffusion models*, 2021.
>
> [3] J. Pidstrigach et al, *Infinite-dimensional diffusion models for function spaces*, 2023.
>
> [4] H. Chung et al, *Diffusion posterior sampling for general noisy inverse problems*, 2023.
>
> [5] Y. Wang et al, *Zero-shot image restoration using denoising diffusion null-space model*, 2022.

---

> > ### Comment · Reviewer_WxYS · 2023-08-19
> > **Thanks for the rebuttal.**
> >
> > Thank the authors' detailed responses, and I increased my score accordingly.

---

### Official Review · Reviewer_hg8t · 2023-07-06

**Soundness:** 4 excellent
**Presentation:** 3 good
**Contribution:** 3 good
**Rating:** 8
**Confidence:** 3

**Summary:**

Score-based diffusion models are successful in solving inverse problems in a finite-dimensional setting, but infinite-dimensional diffusion models needs to be constructed with care, as the definitions for Lebesgue measures and densities become less clear. The authors extends the work of Pidstrigach et al. [33] of unconditional score matching in infinite dimensions to a conditional setting.

**Strengths:**

I recommend a strong accept for the paper because of its theoretical soundness and its approachable presentation in its explanation.

The paper presents a theoretically elegant and principled approach to solving inverse problems in infinite dimensions, as it is guided by Stuart’s principle and does not involve projection to finite vector spaces and discretization when unnecessary. The paper also analyzes a general scenario with prior distributions absolutely continuous w.r.t. the Gaussian measure, and propose analogous results to Pidstrigach et al. [33].

**Weaknesses:**

I cannot identify a specific point of weakness that has to be addressed. One can argue against its simple experiment but I see a proof-of-concept experiment sufficient for this paper.

**Questions:**

I have a few questions regarding the general aspects of Bayesian inference in infinite dimensions, as I am not familiar with the exact formulation.
- Observational model in the paper occupies a finite-dimensional subspace. Is there a scenario where one cannot find an orthonormal basis such that the observation $y$ only spans a finite subspace?
- While I understand Assumption 1 _can_ be satisfied under certain conditions given by the prior measure's Radon-Nikodym derivatives, but does this assumption suffer from finite training data? If we think about extreme settings with only a few training data, the score matching essentially tries to memorize these noiseless data points, causing an automatic violation of the assumption.

**Limitations:**

- Line 15 typo: it should read "extension of ... to the *conditional* setting".
- Minor citation error on Line 72: The seminal score matching paper [17] has Hyvärinen as the sole author.
- Line 293 typo: “… proposition _on_ such set of conditions”

---

> ### Author Rebuttal · Authors · 2023-08-09
>
> We would like to thank the Reviewer for their positive feedback.
>
> We are happy to clarify our manuscript in response to the Reviewer's questions.
>
> 1. **Finite-dimensional observational model.** Indeed, if the number of observations is finite, we may think that the observations
> only span a finite subspace. There are, however, instances where considering infinite-dimensional measurements can prove advantageous. This is particularly relevant when aiming to demonstrate the robustness of theoretical results, such as the asymptotic behavior of sample errors, regardless of the method used to discretize measurements. Such situations often arise when dealing with data being functions observed on a dense array. Andrew Stuart provides an example in [1, Section 3.5]: the inverse problem is determining the initial condition for the heat equation, given noisy observation of the solution at a positive time.
>
> 2. **About Assumption 1.** This is a good remark. Indeed we may deal with a blow up of the score under such conditions and the results  in Section 4 make this explicit in the case with a Gaussian prior.  However, this situation should only occur with noiseless data. As soon as data are noisy, we believe Assumption 1 should be reasonably easy to satisfy.
>
> 3. **Typos.** Thank you for noticing the typos. We will fix them.
>
> **References**
>
> [1] A. Stuart, *Inverse problems: a Bayesian perspective*, Acta Numerica 19 (2010), 451-559.

---

> > ### Comment · Reviewer_hg8t · 2023-08-10
> > **Post-rebuttal comment**
> >
> > I thank the authors for addressing the points I laid out in the review, and maintain the same score assessment for this paper.

---

### Official Review · Reviewer_pjkQ · 2023-07-07

**Soundness:** 3 good
**Presentation:** 3 good
**Contribution:** 2 fair
**Rating:** 6
**Confidence:** 4

**Summary:**

The author extends score based diffusion from finite dimensional processes to separable Hilbert space processes.
They demonstrate that on a non-linear toy data set that the method can work.

**Strengths:**

The paper reads very well and is easy to follow. It is important to study what happens in general separable Hilbert spaces as many algorithms and models breaks down in the infinite dimensional setting. And even though you always do real application in the finite dimensional setting it is still important because if the method works in a separable Hilbert space setting it will not break down when you increase the precision your finite dimensional discretization.



**Weaknesses:**

The numerical experiment is not very convincing. The results for this toy problem are not very impressive.
I suspect that if one runs multiple parallel version of a  Crank–Nicolson algorithm one would get better result.
It also would be interesting to see how the numerical method scale in practice when you increase the resolution of the grid.

**Questions:**

-When you in  section 4. let $(Af)_i=(v_k,f)$ where $v_k$ is a eigenvector, is it not very obvious what you get? I mean the problem is a infinite series of independent processes and the data don't couple the processes. Does it not mean that you are back to the finite dimensional case (since if $v_j \notin A$ then for that $j$ the processes is equivalent to the prior?

-In the application setting I don't get the prior on $X_0$ and when I look in the code it looks one has used $x_0$ as just a line?
- What is the operator $C$ in this example?

very very minor:
Do you really have to write infinite dimensional in each sentence?

**Limitations:**

.

---

> ### Author Rebuttal · Authors · 2023-08-09
>
> We would like to thank the Reviewer for their critical feedback.
>
> We are happy to clarify our manuscript in response to the Reviewer's questions.
>
> 1. **Numerical experiment not very convincing.** We acknowledge your remark, along with those of the other reviewers, regarding the need for improvement in our toy example. In our global response, we have included a PDF that contains additional experiments that will be incorporated in the paper. We have addressed your specific concern by incorporating results indicating the discretization invariance of our method by sampling the toy conditional distribution over grids with varying discretization. These preliminary results anticipate that we are able to sample the bi-modal non-Gaussian conditional distribution (Figure 1 in the attached PDF). Furthermore, we have included a large-dimensional example that involves learning an amortized approximation to the posterior distribution of a wave-equation-based inverse problem over a $256 \times 256$ dimensional unknown, illustrating the applicability of our method to large-scale problems (Figure 2 in the PDF).
>
>    We sincerely hope that these updates could lead to an improvement in your assessment of the paper.
>
> 2. **When in Section 4 you let $(Af)_i = (v_k,f)$ where $v_k$ is a eigenvector, is it not very obvious what you get?** The Gaussian setting of Section 4 makes it possible to carry out explicit and detailed calculations, because it is indeed possible to work $j$ by $j$. The main purpose of Section 4 is to show that the extension of the score-based diffusion models to the conditional setting is not trivial, but possible, in the infinite-dimensional setting. As a byproduct, it also shows that this extension in the finite-dimensional setting also requires some conditions because the blow-up of the score that we exhibit also holds in the finite-dimensional setting with noiseless data. So even the finite-dimensional setting is not obvious~! The singularity in the conditional score as noise vanishes is indeed a well-known phenomenon in the finite-dimensional setting. It has important implications. For a discussion about the difficulties this singularity causes in practice and numerical methods for approximating the score at small times in score-based models in the finite dimensional case, we refer to [1, 2]. As for our paper, we anticipate that Assumption 1 is reasonably easy to satisfy. However, we must be mindful of the possibility of encountering a blow-up of the score under the conditions described by Reviewer hg8t. To address this concern, we will add a remark in our paper.
>
> 3. **$C$ and prior on $x_0$ in the toy example.** Incorporating the suggestions we have received, we have made the decision to extend and improve Section 6. Consequently, we will be updating our implementation. The revised Section 6 will showcase an extended numerical experiment, demonstrating not only the success of our method in approximating a bi-modal non-Gaussian conditional distribution but also its discretization-invariance in practice as the grid resolution is varied (Figure 1 in the PDF included in the global response). Moreover, we will include a new experiment in the appendix, demonstrating the applicability of our method to a large-scale inverse problem in geophysics, specifically linearized seismic imaging via the Born approximation (Figure 2 in the PDF), that involves estimating a $256 \times 256$-dimensional unknown parameter.
>
>    To address your other questions, in the toy example currently provided in the paper, we do not employ any explicit prior information, and in our amortized variational inference approach the prior is implicitly learned during training using the provided dataset of joint samples. We acknowledge that the notation used in Section 6 might have been misleading, as our focus in the current example in the paper is on learning the conditional distribution of $y$ given $x_0$, while in the rest of the paper it was the opposite (we fixed the notation in the PDF included in the global response). As for the operator $C$, we are utilizing the finite-dimensional approximation outlined in [3, Appendix G]. Specifically, we employ [4, Equation 11] as the projected equation.
>
> 3. **Too many "infinite-dimensional" in the paper.** Thank you for the feedback. We will work on reducing the redundancy in the writing.
>
> **References**
>
> [1] D. Kim, S. Shin, K. Song, W. Kang, and I.-C. Moon, *Soft Truncation: A Universal Training Technique of Score-based Diffusion Model for High Precision Score Estimation*, 2022.
>
> [2] T. Dockhorn, A. Vahdat, and K. Kreis, *Score-Based Generative Modeling with Critically-Damped Langevin Diffusion*, 2022.
>
> [3] J. Pidstrigach, Y. Marzouk, S. Reich, and S. Wang, *Infinite-dimensional diffusion models for function spaces*, arXiv:2302.10130.
>
> [4] Y. Song, J. Sohl-Dickstein, D. P. Kingma, A. Kumar, S. Ermon, and B. Poole, *Score-based generative modeling through stochastic differential equations*, ICLR 2021.

---

> > ### Comment · Reviewer_pjkQ · 2023-08-12
> >
> > Thank you for your answer. You have clarified several things for me. I will consider raising my score.

---

### Author Rebuttal · Authors · 2023-08-09

We thank the Reviewers for their valuable and constructive feedback.

Based on your comments, we have taken significant steps to enhance our Section 6. We are now including additional experiments that demonstrate the applicability of our method to large-scale problems and showcase its discretization invariance. **Preliminary results can be found in the attached PDF, which includes relevant figures for your reference.**

In accordance with your suggestions, we have outlined the following improvements:

1. **First Experiment.** Taking advantage of the additional content page for the camera-ready version, if our paper is accepted we will extend Section 6 by incorporating an experiment demonstrating the discretization invariance of our proposed method by sampling a bi-modal non-Gaussian conditional distribution over grids with varying discretizations. In particular, we present results related to sampling the posterior over various discretizations of the $[-3, 3]$ domain. Figure 1 in the attached PDF showcases the outcomes of this experiment by displaying the samples and the marginal conditional distributions at $y=-1, 0, 0.5$ (we used the relation $X_0= a y^2 + \epsilon$). We enhanced robustness against different discretizations by training the score-based model on data residing on nonuniform grids containing $15$ to $50$ grid points. No changes were made to other hyperparameters compared to the original experiment.

2. **Second Experiment.** We will add a new experiment in the appendix to demonstrate the applicability of our method to a large-scale inverse problem in geophysics, specifically linearized seismic imaging via the Born approximation (Figure 2 in the PDF). More specifically, the problem we address involves estimating the short-wavelength component of the Earth's unknown subsurface squared-slowness model (refer to Figure 2a in the uploaded PDF), using measurements collected at the surface. This inverse problem, known as *seismic imaging*, can be formulated as a linear inverse problem by linearizing the nonlinear relationship between recorded data and the squared-slowness model, as governed by the wave equation. In its simplest acoustic form, the linearization around a background squared slowness model (illustrated in Figure 2b of the uploaded PDF) leads to an inverse problem for estimating the true seismic image (depicted in Figure 2a).

   Given the high dimensionality of the observed data, we summarize it by projecting the data back into seismic image space using the adjoint Born operator, as shown in Figure 2c. This process leads to the reverse-time migrated image, which we utilize instead of data to train our conditional score-based model. We carried out the training for $300$ epochs using $4750$ pairs of true seismic images and associated reverse-time migrated images from the 3D Parihaka real dataset, with each image being of size $256 \times 256$. We used a batch size of $128$ and an initial learning rate of $2 \times 10^{-3}$, which decays to $5 \times 10^{-4}$ over the epochs following a power-law rate of $-1/3$.

   Post-training, for a new seismic image (refer to Figure 2a in the uploaded PDF), we simulate seismic data and compute the reverse-time migrated image using the adjoint Born operator. The trained conditional score-based model is then employed to sample from the posterior distribution of the seismic image, given the reverse-time migrated image. We draw $1000$ samples to compute the conditional (posterior) mean estimator, visualized in Figure 2c of the PDF. These samples are also utilized to calculate pointwise standard deviations (Figure 2d) as a measure of uncertainty.

   As anticipated, the pointwise standard deviation highlights areas of high uncertainty, particularly in regions with complex geological structures$-$such as near intricate reflectors and areas with limited illumination (deep and close to boundaries). The regions of significant uncertainty correspond well with challenging-to-image sections of the model. This observation becomes more apparent in Figures 2h and 2i, displaying two vertical profiles with $99$\% confidence intervals (depicted as orange-colored shading), which demonstrate the expected trend of increased uncertainty with depth. Furthermore, we notice that the ground truth (indicated by dashed black lines) largely falls within the confidence intervals for most areas.

   We also observe a strong correlation between the pointwise standard deviation and the error in the conditional mean estimate (Figure 2e), confirming the accuracy of our Bayesian inference method. To prevent bias from strong amplitudes in the estimated image, we present the normalized pointwise standard deviation divided by the envelope of the conditional mean in Figure 2f. This visualization provides an amplitude-independent assessment of uncertainty, highlighting regions of high uncertainty at the onset and offset of reflectors (both shallow and deeper sections). Additionally, the normalized pointwise standard deviation underscores uncertainty in areas of the image where there are discontinuities in the reflectors (indicated by black arrows), potentially indicating the presence of faults.

With these updates and the additional improvements we are implementing based on our responses in the rebuttal, we sincerely hope to have addressed the concerns raised by the Reviewers. If our responses satisfied your concerns, we kindly ask that you consider revisiting your score accordingly. Please let us know if there is anything else that we can do or clarify to enhance the quality of this paper. Once again, thank you for your feedback.

---

### Decision · Program_Chairs · 2023-09-21

**Decision:**

Accept (spotlight)

**Comment:**

The reviewers agree that is paper is an interesting and useful contribution to the literature, especially after the discussions with the authors. Please reread the reviews and your responses carefully, and make the necessary changes in the final submission.